# Calibration-Guided Quantile Regression: Along the Pareto Front of Calibration and Sharpness

## Abstract

Obtaining finite-sample guarantees for predictive models is crucial for settings where decisions have to be made under uncertainty. This has motivated works where models are trained, then recalibrated to yield coverage guarantees. However, doing so often significantly increases model entropy, i.e., it becomes less sharp, making the model less useful. To mitigate this, recent works have increasingly attempted to achieve a balance between good calibration and sharpness. However, these methods often involve deriving completely new, poorly understood loss functions or employing a complex and computationally intensive training pipeline. Moreover, the trade-off between sharpness and calibration is frequently unclear for these methods. In this work, we argue for making the trade-off explicit by choosing the sharpest model subject to some pre-set miscalibration tolerance. To achieve this, we present a simple yet effective approach that combines two established metrics in a novel fashion: we minimize the pinball loss while controlling for calibration using a held-out dataset. Coincidentally, our method motivates a hitherto unexplored analysis: we explicitly compute the Pareto front achieved across methods in terms of sharpness and calibration, and compare performance against this Pareto front. Our approach consistently outperforms various state-of-the-art methods in terms of various Pareto front-related metrics, even though the competing methods are more complex and computationally expensive.

## 1 Introduction

In real-world settings, observations frequently come with some form of aleatoric uncertainty. In such settings, we would ideally like to predict the full statistics of a variable, e.g., to better assess the risk associated with an underlying decision. To address this, various types of stochastic models have been proposed, ranging from Bayesian techniques (MacKay, 2002; Rasmussen & Williams, 2006) to quantile regression (Padilla et al., 2022) and deep ensembles (Lakshminarayanan et al., 2017). However, these methods tend to be miscalibrated when used out of the box, i.e., there is a mismatch between predictive intervals and their respective coverage. This is most frequently addressed by performing some form of post-hoc calibration. Typically, calibration techniques employ a held-out dataset to map model predictions to statistically valid probability distributions (Niculescu-Mizil & Caruana, 2005).

Though calibration has seen increasing interest in recent years (Guo et al., 2017; Kuleshov et al., 2018; Vovk et al., 2020; Roman et al., 2021; Zhao et al., 2021), the underlying methods present several drawbacks. Most methods recalibrate the existing model by some form of post-hoc processing. This can severely distort the model output distribution, leading to unnecessarily underconfident predictions. A related problem is that recalibration approaches often disregard sharpness, i.e., the ability to make predictions with low entropy. This is especially desirable in practice, as high-entropy predictions typically have a significant predictive error, rendering them of little value, even when perfectly calibrated.

To address the loss of sharpness that typically comes with post-hoc calibration, there has been an increasing number of works that attempt to strike a balance between both, e.g., (Kumar et al., 2018; Chung et al., 2021; Kuleshov & Deshpande, 2022; Berta et al., 2025). This is achieved by optimizing

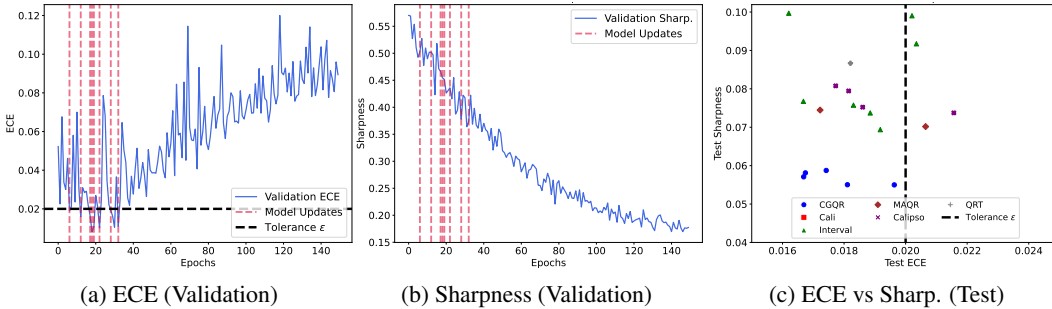

(a) ECE (Validation)    (b) Sharpness (Validation)    (c) ECE vs Sharp. (Test)

Figure 1: Our method aims to find the sharpest model subject to some miscalibration tolerance $\epsilon$. It trains a model using the pinball loss and selects the best one observed during training.(1a) Validation ECE during training. A new candidate optimal model is selected only if the miscalibration error is below $\epsilon$. Instances where the candidate optimal model is updated are highlighted by the red vertical lines. (1b) Sharpness on the validation set during training. The candidate optimal model is updated only if it improves upon the current best sharpness, while its miscalibration error remains below $\epsilon$. (1c) ECE vs sharpness on test data for multiple seeds. Our method stays below the miscalibration tolerance $\epsilon$ and yields the best calibration/sharpness trade-off.

the model while still attempting to achieve perfect calibration (Kumar et al., 2018; Berta et al., 2025) or minimizing a loss function that implicitly trades off calibration and sharpness (Chung et al., 2021; Kuleshov & Deshpande, 2022). However, no method exists that attempts to trade off calibration and sharpness by imposing a maximal miscalibration error. Perhaps somewhat surprisingly, there have been few efforts that map the Pareto front of models that achieve the best sharpness-calibration trade-off.

Motivated by the shortcomings mentioned above, we present Calibration-Guided Quantile Regression (CGQR). Our method achieves an explicit trade-off between calibration and sharpness by allowing miscalibration up to some pre-specified tolerance. This is achieved by training the model using simultaneous quantile regression, which enables an explicit trade-off between calibration and sharpness. Our approach offers finite-sample guarantees similar to those of conventional calibration methods, and demonstrates how the miscalibration error decreases with increasing data size. By performing a thorough analysis of the calibration-sharpness Pareto front, we conclude that the simultaneous pinball loss frequently achieves the Pareto optimal trade-off compared to more computationally expensive training pipelines and losses engineered to achieve a good trade-off.

The remainder of this paper is structured as follows. In Section 2, we review the background relevant to this paper, after which we formally state the problem of maximizing sharpness subject to calibration. Section 3 introduces our method and presents corresponding finite-sample guarantees. We present experimental results, in Section 4, after which we discuss related work, in Section 5. We discuss our method and its limitations in Section 6 and conclude the paper with Section 7.

## 2 BACKGROUND

### 2.1 NOTATION

In this paper, we consider a regression task, where the ground-truth distribution is characterized by its input distribution $F_X(\cdot)$ and conditional output cumulative density function $F(\cdot \mid X)$. We use $q^*(\cdot, \tau)$ and $f(\cdot \mid X)$ to denote the corresponding conditional $\tau$-quantile and probability density functions, respectively. Unless otherwise stated, we use $\mathbb{E}$ and $\mathbb{P}$ to denote the expected value and probability of an event under the joint distribution, i.e., $\mathbb{P} := \mathbb{P}_{x \sim F_X(\cdot), \, y \sim F(\cdot \mid x)}$ and $\mathbb{E} := \mathbb{E}_{x \sim F_X(\cdot), \, y \sim F(\cdot \mid x)}$. For a finite dataset $\mathcal{D}$, we use $\mathbb{E}_{\mathcal{D}} := \mathbb{E}_{(x,y) \sim \mathcal{D}}$ and $\mathbb{P}_{\mathcal{D}} := \mathbb{P}_{(x,y) \sim \mathcal{D}}$ to denote the expected value and probability with respect to $\mathcal{D}$, respectively.

## 2.2 CALIBRATION AND SHARPNESS

Calibration refers to the ability of a model to output distributions that agree with the observed data. While different notions of calibration with varying strengths exist (Zhao et al., 2020), this paper focuses on *average* calibration, which is arguably one of the most studied forms, and is henceforth referred to simply as calibration.

**Definition 2.1** (Calibration). *A model $\hat{q}$ is said to be (average) calibrated if the corresponding predictive quantiles satisfy*

$$\mathbb{P}\left(\hat{q}(X, \tau) \leq Y\right) = \tau, \qquad \forall \, \tau \in [0, 1]. \tag{1}$$

Intuitively, a model is calibrated if the predictive quantile agrees with the observed coverage frequency on average over the ground truth distribution.

Generally, achieving perfect calibration is impossible with a finite sample size. Instead, the best we can settle for is minimizing the expected calibration error (ECE), given by

$$\mathrm{ECE}(\hat{q}) = \int_{\tau=0}^{1} \left|\mathbb{P}\left(\hat{q}(X, \tau) \leq Y\right) - \tau\right| d\tau. \tag{2}$$

Note that, if a single quantile level is of higher importance than others, (2) can be modified straightforwardly to reflect this by employing a weighted integral Kuleshov et al. (2018). We denote the empirical counterpart of ECE on a finite dataset $\mathcal{D}$ by

$$\widehat{\mathrm{ECE}}_{\mathcal{D}}(\hat{q}) = \int_{\tau=0}^{1} \frac{1}{|\mathcal{D}|} \sum_{(x_i, y_i) \in \mathcal{D}} \left|(\hat{q}(x_i, \tau) \leq y_i) - \tau\right| d\tau. \tag{3}$$

A common way of minimizing the ECE of a pre-trained model is by applying recalibration (Gneiting et al., 2007; Kuleshov et al., 2018), where the predicted quantile levels are projected to new levels that correspond to correct average quantiles on a held-out dataset. This procedure effectively yields an empirical ECE (3) of zero over the held-out dataset, and comes with finite-sample guarantees for future ground truth observations (Vovk et al., 2020). However, a drawback of applying recalibration is that it can severely deform the output distribution of the model, leading to the loss of desirable properties obtained during training, e.g., low-entropy output distributions. To contend with this, calibrated models are typically also evaluated by measuring the concentration of the output distributions. This property is known as sharpness.

**Definition 2.2.** *For two models $\hat{q}$ and $\hat{q}'$, we say that $\hat{q}$ has higher sharpness than $\hat{q}'$ if its output distributions have lower entropy on average.*

Hence, sharpness is a model-dependent quantity and does not depend on the conditional ground truth distribution. The gold standard when designing (approximately) calibrated models has become to achieve good sharpness subject to calibration (Gneiting et al., 2007; Kuleshov et al., 2018; Chung et al., 2021). In practice, it is common to measure sharpness using different proxy metrics instead of entropy (Gneiting et al., 2007). In this paper, we evaluate sharpness using the average length of the centered $95\%$ confidence interval:

$$\mathrm{IL}(\hat{q}) = \mathbb{E}\left[\hat{q}(x, 0.975) - \hat{q}(x, 0.025)\right]. \tag{4}$$

This metric is particularly interesting in practice, as it measures the tightness of confidence intervals, a tool commonly used to assess risk.

## 2.3 QUANTILE REGRESSION

Quantile regression (Koenker & Bassett, 1978) is a well-established tool for modeling quantile functions. It trains a model by applying the pinball loss, also commonly known as the quantile loss or check score, which is given by

$$l_\tau(\hat{q}, \tau) = \begin{cases} \tau\left(y - \hat{q}(x, \tau)\right) & \text{if } y \geq \hat{q}(x, \tau) \\ (1 - \tau)\left(\hat{q}(x, \tau) - y\right) & \text{if } y < \hat{q}(x, \tau). \end{cases} \tag{5}$$

Intuitively, the pinball loss penalizes the weighted $L^1$ distance in an asymmetric fashion, where the weights correspond to the desired quantile $\tau$ and $1 - \tau$. In this paper, we train a model to have a small pinball loss for any quantile level $\tau$. To this end, we marginalize over $\tau$, obtaining the loss

$$L(\hat{q}) = \int_{\tau=0}^{1} l_\tau(\hat{q}, \tau). \tag{6}$$

Training a model by minimizing (6) is commonly known as *simultaneous quantile regression* (Bondell et al., 2010), and comes with many benefits. It accounts for multimodality, heteroskedasticy, and asymmetry in the data distribution by modeling each quantile separately. Moreover, if $\hat{q}$ is continuous in $\tau$, minimizing (6) implicitly regularizes the difference between quantiles of different levels $\tau$ (Schnabel & Eilers, 2013). A further advantage of training a model in this manner (6) corresponds to a proper scoring rule (Gneiting & Raftery, 2007), meaning that the ground truth distribution minimizes this loss.

Importantly, (6) implicitly trades off calibration and sharpness. This can be best seen by analyzing the pinball loss for a single quantile level $\tau$ and small miscalibration errors as follows: Assume $\hat{q}(X, \tau) > q(X, \tau)$ without loss of generality. The expected pinball loss then allows a partition into a model-dependent and a model-free component:

$$\mathbb{E}[l_\tau(q(X, \tau), Y)] = \underbrace{\mathbb{E}\left[ \int_{q(X,\tau)}^{\hat{q}(X,\tau)} \left( F(u \mid X) - \tau \right) du \right]}_{\text{Model-dependent}} + \underbrace{\mathbb{E}[l_\tau(q(X, \tau), Y)]}_{\text{Irreducible}}. \tag{7}$$

By applying a first-order approximation to the model-dependent term (see Appendix A), we obtain

$$\mathbb{E}\left[ \int_{q(X,\tau)}^{\hat{q}(X,\tau)} \left( F(u \mid X) - \tau \right) du \right] \approx \mathbb{E}\left[ \frac{(F(\hat{q}(X, \tau) \mid X) - \tau)^2}{f(q(X, \tau)+ \mid X)} \right]. \tag{8}$$

The numerator $(F(\hat{q}(X) \mid X) - \tau)^2$ is the square of the conditional calibration error. It corresponds to a much stronger notion of calibration than average calibration, in that, if minimized, it agrees perfectly with the ground truth. It is related to the ECE (2) as

$$\text{ECE}(\hat{q}) = \int_{\tau=0}^{1} \mathbb{E}\left[ F(\hat{q}(X) \mid X) - \tau \right] d\tau \tag{9}$$

In other words, if the expected pinball loss is minimized for all $\tau$, then the model is perfectly calibrated. However, if $F(\hat{q}(X) \mid X) - \tau \neq 0$, the ECE of the model is potentially nonzero. In this case, the loss attempts to simultaneously maximize the denominator $f(q(X, \tau)+ \mid X)$, which in turn encourages the quantiles to cluster around areas of high likelihood, mimicking the entropy of the ground truth, and discouraging unnecessarily high entropy. As we show in Section 4, this trade-off becomes apparent when we explicitly try to reach a good trade-off between calibration and sharpness.

## 2.4 PARETO FRONT EVALUATION

In this paper, we aim to analyze the trade-off between calibration and sharpness, two concurrent objectives in optimization. To do so, we resort to computing the Pareto front across models.

Let $\mathcal{M} = \{\hat{q}_1, \ldots, \hat{q}_N\}$ be a set of models, and let $\text{ECE}_i := \text{ECE}(\hat{q}_i)$ and $\text{IL}_i := \text{IL}(\hat{q}_i)$ denote their respective calibration and sharpness values. The corresponding Pareto front is the subset of points in calibration and sharpness space where no point is dominated by another in both calibration and sharpness metrics simultaneously. Formally, this is given by the models $i$ that satisfy

$$\mathcal{P}(\mathcal{M}) = \left\{ (\text{ECE}_i, \text{IL}_i) \; \middle| \; \text{ECE}_i < \text{ECE}_j \vee \text{IL}_i < \text{IL}_j \;\; \forall j, \;\; i \neq j \right\}. \tag{10}$$

For any subset $\hat{\mathcal{P}} \subset \mathcal{M}$, there exist several metrics that measure how well it approximates the true Pareto front $\mathcal{P}$ (Audet et al., 2021). In this paper, we consider three common indicators: the hypervolume (HV), the generational distance (GD), the inverse generational distance (IGD). We describe these metrics in the following.

The HV measures the area between a reference point and the approximated Pareto front $\hat{\mathcal{P}}$ generated by each model in a subset corresponding to a specific loss. In this paper, we employ the maximal calibration error and interval length observed across models as a reference point. Assuming that $\text{ECE}_j < \text{ECE}_{j+1}$ for all $\text{ECE}_j \in \hat{\mathcal{P}}$, the hypervolume is given by

$$\text{HV}(\hat{\mathcal{P}}) = \sum_{i=1}^{|\hat{\mathcal{P}}|} (\text{IL}^{\max} - \text{IL}_i)\left(\text{ECE}_{i+1} - \text{ECE}_i\right), \qquad \text{with } \text{ECE}_{|\hat{\mathcal{P}}|+1} := \text{ECE}^{\max}. \tag{11}$$

Intuitively, the hypervolume measures the convergence of an approximate Pareto front to the true one, as well as the diversity of the approximation. The closer the ratio $\text{HV}(\hat{\mathcal{P}})/\text{HV}(\mathcal{P})$ is to one, the better the approximation.

A drawback of the HV is that it does not measure the uniformity of the approximation $\hat{\mathcal{P}}$, i.e., how spread out it is compared to the true Pareto front. To measure this, we additionally report the GD and IGD metrics, given by

$$\text{GD}(\hat{\mathcal{P}}, \mathcal{P}) = \left(\frac{1}{|\hat{\mathcal{P}}|} \sum_{z \in \hat{\mathcal{P}}} \min_{u \in \mathcal{P}} \|z - u\|_2^2\right)^{\frac{1}{2}}, \qquad \text{IGD}(\hat{\mathcal{P}}, \mathcal{P}) = \left(\frac{1}{|\mathcal{P}|} \sum_{z \in \mathcal{P}} \min_{u \in \hat{\mathcal{P}}} \|z - u\|_2^2\right)^{\frac{1}{2}},$$

Unlike the HV distance, these metrics measure how spread out an approximation of $\hat{\mathcal{P}}$ is compared to $\mathcal{P}$.

## 3 PROBLEM STATEMENT AND METHOD

### 3.1 PROBLEM STATEMENT

In this paper, we consider the problem of achieving an *explicit* trade-off between calibration and sharpness for various permissible levels of miscalibration. More specifically, given a model architecture $q_\theta$ and some miscalibration tolerance $\epsilon$, we seek to find model parameters $\theta^*$ that (approximately) satisfy

$$\theta^* = \arg\min_\theta \text{IL}(q_\theta), \quad \text{s.t.} \quad \text{ECE}(q_\theta) \leq \epsilon. \tag{12}$$

Note that if we vary $\epsilon$ from 0 to 1, the solutions of (12) lie on the Pareto front of calibration and sharpness values. Hence, our goal is aligned with approximating the Pareto front that quantifies the best trade-off between calibration and sharpness.

### 3.2 CALIBRATION-GUIDED QUANTILE REGRESSION

We now describe our approach, which aims to minimize sharpness subject to some miscalibration error $\epsilon$, as shown in (12). To find the (approximately) optimal parameters $\theta^*$, we minimize the pinball loss, allowing us to leverage the calibration-sharpness trade-off described in Section 2.3. As shown later, in Section 4, this trade-off is considerably superior to that obtained by other state-of-the-art approaches for a broad spectrum of miscalibration tolerances $\epsilon$.

Our approach requires the specification of a miscalibration tolerance $\epsilon$, to be decided by practitioners. We first split the data set into a training set $\mathcal{D}_{\text{tr}}$ and a validation set $\mathcal{D}_{\text{val}}$. The training data set $\mathcal{D}_{\text{tr}}$ is used to optimize the candidate model using the pinball loss. The validation set $\mathcal{D}_{\text{val}}$, by contrast, is used exclusively to assess calibration and sharpness during training, which we then use to discriminate among candidate models. Given some initial model parameters $\theta$, we then initialize the best model parameters as $\theta^* = \theta$. Given $\epsilon$, we then perform multiple gradient descent steps to optimize the pinball loss $\mathbb{E}_{\mathcal{D}_{\text{tr}}}[\int_{\tau=0}^1 l_\tau(\hat{q}_\theta, Y) d\tau]$. This is achieved by sampling a batch of quantile levels from the uniform distribution between 0 and 1, then performing gradient descent on the sum of the resulting pinball losses. However, unlike in standard training, after every iteration, we check if the current model $\hat{q}_\theta$ respects the miscalibration tolerance on the validation dataset $\mathcal{D}_{\text{val}}$, i.e., $\widehat{\text{ECE}}_{\mathcal{D}_{\text{val}}}(\hat{q}_\theta) \leq \epsilon$. If this is not the case, we proceed to do another optimization step. Otherwise, we check if the average interval length is smaller than for the best model stored, i.e., $\text{IL}_{\mathcal{D}_{\text{val}}}(\hat{q}_\theta) < \text{IL}_{\mathcal{D}_{\text{val}}}(\hat{q}_{\theta^*})$. If this is the case, we set the best parameters to $\theta^* = \theta$. This procedure is repeated until convergence has been reached or a pre-specified number of steps has been taken. This is outlined in Algorithm 1.

---

**Algorithm 1** Calibration-Guided Quantile Regression

---

**Input:** Parametric model $\hat{q}_\theta$, data $\mathcal{D}$, miscalibration tolerance $\epsilon$

1: Split $\mathcal{D}$ into training data $\mathcal{D}_{tr}$ and validation data $\mathcal{D}_{val}$
2: Initialize $\theta^* = \theta$
3: **while** not converged **do**
4:     Update $\theta$ by performing gradient descent on $\mathbb{E}_{\mathcal{D}_{tr}}\left[\int_{\tau=0}^1 l_\tau(\hat{q}_\theta, Y)d\tau\right]$
5:     **if** $\widehat{\text{ECE}}_{\mathcal{D}_{val}}(\hat{q}_\theta) \leq \epsilon$ and $\text{IL}_{\mathcal{D}_{val}}(\hat{q}_\theta) < \text{IL}_{\mathcal{D}_{val}}(\hat{q}_{\theta^*})$ **then**
6:         Set $\theta^* = \theta$
7:     **end if**
8: **end while**
9: **return** $\hat{q}_{\theta^*}$

---

### 3.3 THEORETICAL RESULTS

Our algorithm provides finite-sample guarantees that can be computed straightforwardly based on the miscalibration tolerance $\epsilon$, the validation data set size $|\mathcal{D}_{val}|$, and the number of models $m(\epsilon)$ with miscalibration error below $\epsilon$ observed during training. To this end, we require the following assumption on the dataset distribution.

**Assumption 3.1.** *For any finite number of samples $X_i, Y_i$, from the ground truth distribution, $Y_i$ are exchangeable random variables and almost surely distinct.*

Assumption 3.1 is relatively mild and is satisfied, e.g., by settings where the output is iid and perturbed by Gaussian noise.

**Theorem 3.2.** *Let Assumption 3.1 hold. For some tolerance parameter $\epsilon$, let $\theta^*$ be the optimal model parameters obtained with Algorithm 1. Furthermore, let $m(\epsilon)$ be the total number of parameters $\theta$ observed during training that satisfy*

$$\widehat{ECE}_{\mathcal{D}_{val}}(q_\theta) \leq \epsilon.$$

*Then, for any $0 < \gamma < 1$, with probability at least $1 - 2m(\epsilon)\exp\left(-2\gamma^2|\mathcal{D}_{val}|\right)$,*

$$ECE(\hat{q}_{\theta^*}) \leq \epsilon + \frac{1}{|\mathcal{D}_{val}|} + \gamma. \tag{13}$$

Theorem 3.2 indicates that there is an interplay between the number of models $m(\epsilon)$ out of which we pick $\theta^*$ and the size of the validation data set $\mathcal{D}_{val}$. If Algorithm 1 encounters more models $m(\epsilon)$ that satisfy the miscalibration tolerance, the sharpness on the test set will potentially improve while calibration will potentially suffer. However, this can be offset by having a larger validation set $\mathcal{D}_{val}$, as it yields a more robust calibration guarantee, allowing for a greater number of models $m(\epsilon)$ to be compared. We note that $m(\epsilon)$ can be reduced artificially, e.g., by randomly discarding a model without comparing it, though this comes at the expense of model sharpness.

## 4 EXPERIMENTS

We evaluate the performance of our approach on various regression tasks from open-source repositories and a real-world nuclear fusion prediction problem. Next, we describe the regression tasks, followed by an outline of the training procedure and baseline comparisons.

**Datasets from Open-Source Repositories** We consider the regression tasks associated with the following datasets, available from the UCI, OpenML, and Kaggle repositories: yacht (Gerritsma et al., 1981), Boston house prices (Harrison & Rubinfeld, 1978), wine (Aeberhard et al., 1991), concrete (Yeh, 1998), kin8nm (Dua & Graff, 2017a), naval (Coraddu et al., 2016), power (Dua & Graff, 2017b), protein (Rana, 2013), diamonds (Wickham et al., 2025), Facebook comment volume II (Singh, 2015), and elevator (Axenie & Bortoli, 2020). The dataset sizes range from $|\mathcal{D}| = 307$ (naval) all the way to $|\mathcal{D}| = 112,000$ (elevator).

Table 1: Pareto front metrics in calibration and sharpness space. We report the IGD, GD, and normalized HV, as described in Subsection 2.4, averaged across all datasets. Lower is better for GD and IGD, higher is better for HV. Our method (CGQR) is evaluated, along with Cali, Interval and Model Agnostic Quantile Regression (MAQR) (Chung et al., 2021), CaliPSo (Capone et al., 2025), and QRT (Dheur & Ben taieb, 2024). Lower is better for all metrics. The best results across methods are highlighted in bold.

|  | CGQR | CALI | INTERVAL | MAQR | CALIPSO | QRT |
|---|---|---|---|---|---|---|
| IGD ($\downarrow$) | **0.28±0.04** | **0.35±0.04** | 0.49±0.04 | 0.62±0.04 | 0.51±0.04 | 0.49±0.07 |
| GD ($\downarrow$) | **0.20±0.02** | 0.35±0.03 | 0.38±0.06 | 0.59±0.06 | 0.30±0.05 | 0.63±0.21 |
| HV ($\uparrow$) | **0.92 ± 0.01** | 0.82 ± 0.01 | 0.73 ± 0.01 | 0.51 ± 0.02 | 0.67 ± 0.02 | 0.78 ± 0.02 |

**Nuclear Fusion Dataset**  We additionally consider a nuclear fusion regression task, which corresponds to a dataset obtained from the DIII-D tokamak (Holcomb & for the DIII-D Team, 2024). The plasma dynamics in tokamaks are notoriously difficult to predict in general due to their high degree of stochasticity and poorly understood physics, eliciting an increased interest in using machine learning techniques for prediction (Char et al., 2023; Seo et al., 2024; Sonker et al., 2025). The dataset we use corresponds to a one-step prediction model. It consists of approximately 1.7 million input-output pairs. Each input has $d = 35$ dimensions, consisting of various states and actuators (see Appendix D). The target variable is the principal component of the rotation profile at the next time step. Predicting the distribution of the rotation profile is generally useful, as controlling helps improve the stability of the plasma (Richner et al., 2024).

### 4.1 SETUP AND BASELINES

For all tasks, we consider miscalibration errors of $\epsilon \in (0, 0.15)$. As a model $\hat{q}$, we employ an 8-layer neural network with 256 hidden dimensions and residual connections. Although we experimented with smaller architectures, we observed that they consistently lacked expressivity, yielding less competitive Pareto fronts (see Appendix C). We additionally compare our approach against five different baselines: the calibration loss (Cali), interval loss (Interval), and model-agnostic quantile regression (MAQR) from Chung et al. (2021), quantile recalibration training (QRT) (Dheur & Ben taieb, 2024), and calibrated predictive models with sharpness as loss function (CaliPSo) (Capone et al., 2025). The Cali and Interval baselines minimize surrogate losses that balance calibration and sharpness. MAQR trains a neural network to predict the mean of the data, then trains a quantile model on the residuals binned by proximity. QRT employs a smoothened calibration procedure, then differentiates through calibration during training, which uses a negative log likelihood loss. Similarly, CaliPSo constructs a differentiable model that stays calibrated on the training data, then attempts to maximize its sharpness. We train each model over 1000 epochs. A detailed description of the baselines is given in Appendix E.

We employ the same architecture across all methods, with the difference that, for QRT, the network outputs two quantities (mean and variance) instead of one. To assess how closely each method approximates the Pareto front, we store models that achieve the highest sharpness subject to a miscalibration constraint on the validation data, following a similar approach to ours. For the open-source datasets, we employ a train/validation/test data split of $80/10/10$, except for the Facebook and elevator datasets, where we employ a split of $25/25/50$ to alleviate computational costs. For the fusion dataset, we sample only $|\mathcal{D}| = 200,000$ points from the full data set per seed, and carry out experiments using a $25/25/50$. We run 5 different seeds per dataset except for the Facebook, elevator, and fusion datasets, where we use 3 seeds.

For every seed, we compute the set of Pareto-optimal solutions by pooling the models obtained by each method and discarding those that are dominated in both calibration and sharpness. We then quantify how well each method approximates the Pareto front using the metrics described in Subsection 2.4.

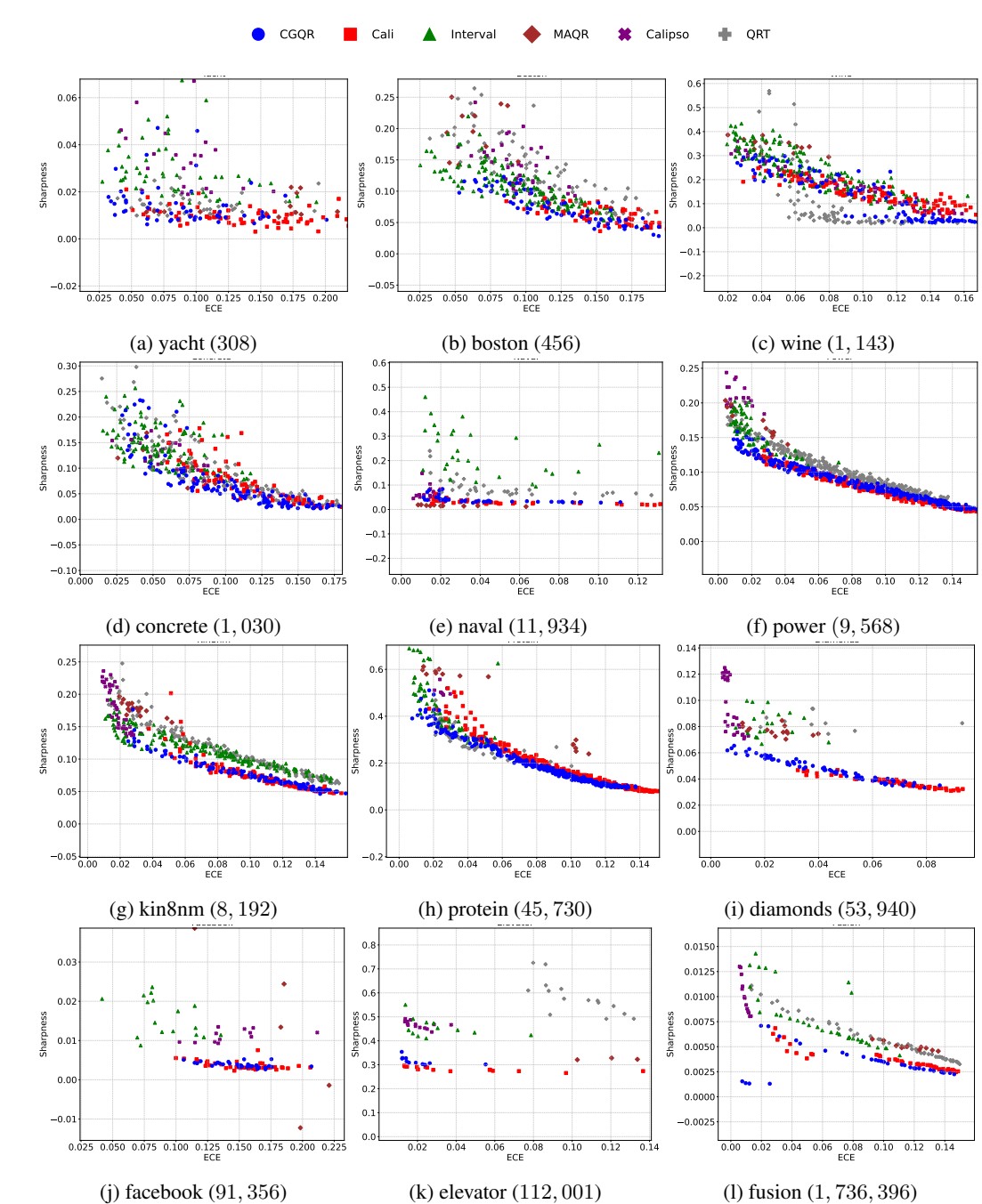

Figure 2: Test time performance of methods in terms of ECE (x-axis) vs Sharpness (y-axis) for different datasets. The number in Lower is better for both metrics. The numbers in brackets indicate the dataset size. Our approach is consistently close to the Pareto front, regardless of dataset size.

## 4.2 RESULTS

The ECE and sharpness of the different models across all seeds is shown in Fig. 5. The summary of the Pareto front metrics is shown in Table 1. The full table is in Appendix G. Our approach consistently achieves the best or close to best hypervolume, indicating that it encloses a space similar to that of the Pareto front. The only method that performs comparably, albeit slightly worse, is Cali, which trains the model by minimizing a weighted sum that penalizes poor ECE and sharpness.

Our approach also consistently reaches a remarkably high ECE. This is due to how the pinball loss weights calibration and sharpness, as discussed in Section 2.3. To reach even higher calibration, an isotonic recalibration approach can be employed (Kuleshov et al., 2018). However, we observed that this often only marginally improves calibration, while markedly deteriorating sharpness (see Appendix F).

## 5 RELATED WORK

### 5.1 CALIBRATED MODELS

Calibration was initially developed in the context of classification, where it was most commonly achieved using Platt scaling (Platt et al., 1999) and isotonic regression (Niculescu-Mizil & Caruana, 2005). More recently, different variants of logistic regression have been used to obtain calibration, such as temperature scaling (Guo et al., 2017), beta calibration (Kull et al., 2017), and Dirichlet calibration (Kull et al., 2019). In the context of regression, Kuleshov et al. (2018) presented an isotonic regression-based calibration approach for neural networks. A general overview of basic recalibration methods plus theoretical guarantees is provided in Marx et al. (2022).

### 5.2 CALIBRATION AND SHARPNESS TRADE-OFFS

To mitigate the loss in sharpness that comes with pure calibration, various works attempt to balance calibration and sharpness. Typically, a trade-off can be obtained by minimizing some form of proper scoring rule (Gneiting & Raftery, 2007). The works of Song et al. (2019); Kuleshov & Deshpande (2022) attempt to achieve distribution calibration, corresponding to minimizing the average pinball loss. In Chung et al. (2021), the authors propose different losses that trade off calibration and sharpness. Kumar et al. (2018) proposes a differentiable calibration metric, which is optimized jointly with the negative log likelihood during training. In Dheur & Ben taieb (2024), the authors propose a differentiable recalibration approach for regression, which allows them to enforce calibration during training and minimize the negative log-likelihood. In a similar vein, Capone et al. (2024) exploits properties of kernel models to formulate a flexible calibration approach that can be optimized for sharpness.

## 6 DISCUSSION

Even though our approach provides finite-sample statistical guarantees, the corresponding bounds are not tight, as they are obtained by applying the union bound (see Appendix B). A way to potentially improve them would be to closely look at the correlation between all the candidate models considered when picking the optimal one. Theorem 3.2 applies to a setting where we discriminate between any set of models, regardless of how they are obtained. Hence, it can potentially be used to discriminate among models that are obtained through different training procedures. This allows us to train multiple models in parallel, e.g., using different loss functions, and then pick the best one according to the procedure detailed in Algorithm 1. A potential downside of the model considered in Section 4 is that it does not explicitly enforce non-crossing quantiles. While this is not uncommon (Chung et al., 2021; Kuleshov & Deshpande, 2022), it may be problematic in settings where the interpretability of the model is important. This can be mitigated by employing different techniques that rigorously enforce this requirement (Bondell et al., 2010), though performance may otherwise suffer.

## 7 CONCLUSION

We have presented calibration-guided quantile regression, a simple technique that minimizes the pinball loss to find the sharpest model subject to some miscalibration error. Our method is straightforward to implement and combines two well-established quantities in a novel manner. We have carried out a thorough Pareto analysis of the calibration/sharpness trade-off, which shows that our approach achieves a better trade-off than more involved and computationally expensive state-of-the-art approaches. Our work also indicates potential exciting future directions, such as choosing a model from a larger pool of candidates, and using Pareto fronts to devise new loss functions.

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

## A    APPROXIMATION OF EXPECTED PINBALL LOSS

By employing the change of variables $z = F(u \mid X) - \tau$, the model-dependent component can be reformulated as

$$\mathbb{E}\left[\int_{q(X,\tau)}^{\hat{q}(X,\tau)} \left(F(u \mid X) - \tau\right) du\right] = \mathbb{E}\left[\int_{z=0}^{F(\hat{q}(X)|X)-\tau} \frac{z}{f\left(q(X,\tau+z) \mid X\right)} dz\right] \quad (14)$$

Assuming $f\left(q(X,\tau+z) \mid X\right)$ is differentiable in $z$, a first-order Taylor expansion then yields

$$
\begin{aligned}
&\mathbb{E}\left[\int_{z=0}^{F(\hat{q}(X)|X)-\tau} \frac{z}{f\left(q(X,\tau+z) \mid X\right)} dz\right] \\
=&\mathbb{E}\left[\int_{z=0}^{F(\hat{q}(X)|X)-\tau} 0 + \frac{1}{f\left(q(X,\tau)+ \mid X\right)} z + \mathcal{O}(z^2)\, dz\right] \\
\approx&\mathbb{E}\left[\int_{z=0}^{F(\hat{q}(X)|X)-\tau} \frac{z}{f\left(q(X,\tau)+ \mid X\right)} dz\right] \\
=&\mathbb{E}\left[\frac{\left(F(\hat{q}(X) \mid X) - \tau\right)^2}{f\left(q(X,\tau)+ \mid X\right)}\right].
\end{aligned}
\quad (15)
$$

## B    PROOF OF THEOREM 3.2

To prove our result, we employ the following result, which is due to Romano et al. (2019).

**Lemma B.1** (Romano et al. (2019), Lemma 2). *Suppose $Z_1, \ldots, Z_n, Z_{n+1}$ are exchangeable random variables and almost surely distinct. Let $Z_{(\lceil(n+1)\alpha\rceil)}$ denote the $\lceil(n+1)\alpha\rceil$-th smallest value in $Z_1, \ldots, Z_n$. Then, for any $\alpha \in (0,1)$,*

$$\alpha \leq \mathbb{P}\left[Z_{n+1} \leq Z_{(\lceil(n+1)\alpha\rceil)}\right] \leq \alpha + \frac{1}{n}. \quad (16)$$

As a direct consequence, the empirical miscalibration for a single quantile level $\tau$ and model $\hat{q}$ is accurate up to an additive constant of $\frac{1}{|\mathcal{D}_{\text{val}}|}$:

**Lemma B.2.** *Consider a model $\hat{q}$ and the random variables $Z_1, \ldots, Z_{|\mathcal{D}_{val}|}, Z_{|\mathcal{D}_{val}|+1}$ defined as*

$$Z_i := \hat{q}(X_i,\tau) - Y_i.$$

*Furthermore, let*

$$e_\tau = \tau - \frac{1}{|\mathcal{D}_{val}|} \sum_{i=1}^{|\mathcal{D}_{val}|} \mathbb{1}(Z_i \leq 0) \quad (17)$$

*be the signed miscalibration error for a specific quantile $\tau$. Then,*

$$-e_\tau \leq \mathbb{E}[\mathbb{1}(Z_{n+1} \leq 0) - \tau] \leq -e_\tau + \frac{1}{|\mathcal{D}_{val}|}. \quad (18)$$

*Proof.* We first consider the case where $\tau - e_\tau \in (0,1)$, and then handle the edge cases separately.

By definition of $e_\tau$, exactly $|\mathcal{D}_{\text{val}}|(\tau - e_\tau)$ of the $Z_i$ are nonpositive, so

$$Z_{(|\mathcal{D}_{\text{val}}|(\tau-e_\tau))} \leq 0 < Z_{(|\mathcal{D}_{\text{val}}|(\tau-e_\tau)+1)}.$$

For the upper bound, since $Z_{(|\mathcal{D}_{\text{val}}|(\tau-e_\tau)+1)} \leq Z_{(\lceil(|\mathcal{D}_{\text{val}}|+1)(\tau-e_\tau)\rceil)}$, Lemma B.1 gives

$$\mathbb{P}(Z_{n+1} \leq 0) \leq \mathbb{P}\left(Z_{n+1} \leq Z_{(\lceil(|\mathcal{D}_{\text{val}}|+1)(\tau-e_\tau)\rceil)}\right) \leq (\tau - e_\tau) + \frac{1}{|\mathcal{D}_{\text{val}}|}.$$

Subtracting $\tau$ yields

$$\mathbb{E}[\mathbb{1}(Z_{n+1} \leq 0) - \tau] \leq -e_\tau + \frac{1}{|\mathcal{D}_{\text{val}}|}.$$

For the lower bound, note that $Z_{(|\mathcal{D}_{\text{val}}|(\tau - e_\tau))} \leq 0$. Applying Lemma B.1 with $\alpha = \tau - e_\tau$ gives

$$\mathbb{P}(Z_{n+1} \leq 0) \geq \mathbb{P}\big(Z_{n+1} \leq Z_{(|\mathcal{D}_{\text{val}}|(\tau - e_\tau))}\big) \geq \tau - e_\tau.$$

Subtracting $\tau$ yields

$$\mathbb{E}[\mathbb{1}(Z_{n+1} \leq 0) - \tau] \geq -e_\tau.$$

Now, if $\tau - e_\tau = 0$, we have that $Z_i > 0$ for all $i$, hence, for any $\alpha > 0$,

$$\mathbb{P}\left[Z_{n+1} \leq 0\right] \leq \mathbb{P}\left[Z_{n+1} \leq Z_{\lceil |\mathcal{D}_{\text{val}}| \alpha \rceil}\right] = \mathbb{P}\left[Z_{n+1} \leq Z_1\right] \leq \alpha + \frac{1}{|\mathcal{D}_{\text{val}}|}. \tag{19}$$

Since this holds for any $\alpha > 0$, it must hold for the limit as $\alpha \to 0$, i.e., $\mathbb{P}\left[Z_{n+1} \leq 0\right] \leq \frac{1}{|\mathcal{D}_{\text{val}}|}$. The same argument holds for $\tau - e_\tau = 1$.

$\square$

By applying the union bound, we can obtain a result similar to Lemma B.2 for multiple models jointly:

**Lemma B.3.** *Consider $m$ models $\hat{q}_{\theta_1}, \ldots, \hat{q}_{\theta_m}$ and the random variables $Z_i^{(j)} := \hat{q}_{\theta_j}(X_i, \tau) - Y_i$, $j = 1, \ldots, m$, $i = 1, \ldots, n$. Furthermore, let*

$$e_\tau^{(j)} = \tau - \frac{1}{|\mathcal{D}_{val}|} \sum_{i=1}^{|\mathcal{D}_{val}|} \mathbb{1}(Z_i^{(j)} \leq 0) \tag{20}$$

*be the signed miscalibration error for a specific quantile $\tau$ and model $j$, and assume $\max_j e_\tau^{(j)} \leq \epsilon$. Then, for any $0 < \gamma < 1$, with probability at least $1 - 2m \exp\left(-2\gamma^2 |\mathcal{D}_{val}|\right)$,*

$$\max_j \left| \mathbb{E}\left[ \mathbb{1}\left( Z_{n+1}^{(j)} \leq 0 \right) - \tau \right] \right| \leq \epsilon + \frac{1}{|\mathcal{D}_{val}|} + \gamma \tag{21}$$

*Proof.* We know that, for every $j$,

$$\mathbb{E}\left[ \mathbb{1}\left( Z_{n+1}^{(j)} \leq 0 \right) - \tau \right] := \Delta_j \in \left[ -e_\tau^{(j)} - \frac{1}{|\mathcal{D}_{\text{val}}|}, -e_\tau^{(j)} + \frac{1}{|\mathcal{D}_{\text{val}}|} \right]$$

From the union bound and Hoeffding's inequality, we have

$$\mathbb{P}\left[ \exists j : \left| \Delta_j - \frac{1}{|\mathcal{D}_{\text{val}}|} \sum_{i=1}^{|\mathcal{D}_{\text{val}}|} \mathbb{1}(Z_i^{(j)} \leq 0) + \tau \right| \geq \gamma \right]$$

$$\leq \sum_{j=1}^m \mathbb{P}\left[ \left| \Delta_j - \frac{1}{|\mathcal{D}_{\text{val}}|} \sum_{i=1}^{|\mathcal{D}_{\text{val}}|} \mathbb{1}(Z_i^{(j)} \leq 0) + \tau \right| \geq \gamma \right] \tag{22}$$

$$\leq 2m \exp\left(-2\gamma^2 |\mathcal{D}_{\text{val}}|\right)$$

Hence, with probability at least $1 - 2m \exp\left(-2\gamma^2 |\mathcal{D}_{\text{val}}|\right)$,

$$\max_j \left| \mathbb{E}\left[ \mathbb{1}\left( Z_{n+1}^{(j)} \leq 0 \right) - \tau \right] \right| \leq \max_j e_\tau^{(j)} + \frac{1}{|\mathcal{D}_{\text{val}}|} + \gamma. \tag{23}$$

The result then follows from $\max_j e_\tau^{(j)} \leq \epsilon$.

$\square$

A direct consequence of Lemma B.3 is that we bound the miscalibration error on all models jointly with high probability based on the empirical ECE:

**Lemma B.4.** *Consider $m$ models $\hat{q}_{\theta_1}, \ldots, \hat{q}_{\theta_m}$ and let their empirical miscalibration errors satisfy $\widehat{ECE}_{\mathcal{D}_{val}}(\hat{q}_{\theta_i}) \leq \epsilon$. Then, for any $0 < \gamma < 1$, with probability at least $1 - 2m \exp\left(-2\gamma^2 |\mathcal{D}_{val}|\right)$, their miscalibration errors satisfy*

$$ECE(\hat{q}_{\theta_i}) \leq \epsilon + \frac{1}{|\mathcal{D}_{val}|} + \gamma, \quad \forall j = 1, \ldots, m. \tag{24}$$

*Proof.* From (20) and the definition of empirical miscalibration error, we have

$$\epsilon \geq \int_{\tau=0}^{1} |e_\tau^{(j)}| d\tau. \tag{25}$$

Furthermore, from Lemma B.3, we have that

$$\max_j \text{ECE}(\hat{q}_{\theta_j}) = \max_j \int_{\tau=0}^{1} \left| \mathbb{P}\left[ Z_{n+1}^{(j)} \leq 0 \right] - \tau \right| d\tau$$

$$\leq \max_j \int_{\tau=0}^{1} \left( |e_\tau^{(j)}| + \frac{1}{|\mathcal{D}_{\text{val}}|} + \gamma \right) d\tau \leq \epsilon + \frac{1}{|\mathcal{D}_{\text{val}}|} + \gamma. \tag{26}$$

□

*Proof of Theorem 3.2.* Algorithm 1 discards all but $m(\epsilon)$ models and picks the one with the lowest empirical sharpness among them. Hence, the probability that the chosen model $\hat{q}_{\theta^*}$ satisfies $\text{ECE}(\hat{q}_\theta^*) \leq \epsilon + \frac{1}{|\mathcal{D}_{\text{val}}|} + \gamma$ is upper-bounded by the probability that all $m(\epsilon)$ models satisfy this property jointly. The result then follows from Lemma B.4.

□

# C  PARETO FRONTS ACROSS MODELS

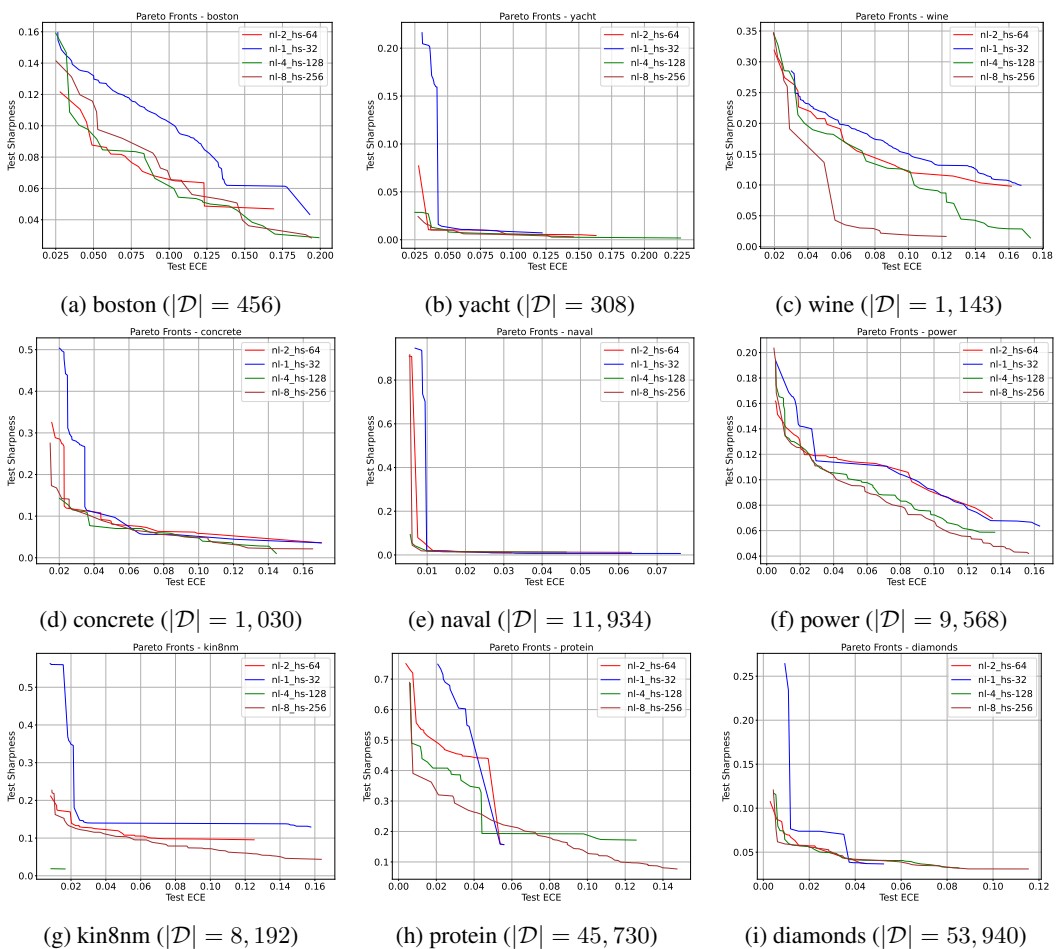

Figure 3: Pareto fronts for different model sizes for each dataset. Red is a single-layer NN with 32 hidden dimensions, blue is a two-layer NN with 64 hidden dimensions, green is a four-layer NN with 128 hidden dimensions, brown is an eight-layer NN with 264 hidden dimensions. The latter is identical to that used in the experimental section of the paper.

## D  NUCLEAR FUSION DATASET

The input variables for the nuclear fusion dataset are as follows:

betan_EFIT01, dssdenest, li_EFIT01, vloop, temp_component1, temp_component2, temp_component3, temp_component4, itemp_component1, itemp_component2, itemp_component3, itemp_component4, dens_component1, dens_component2, dens_component3, dens_component4, rotation_component1, rotation_component2, rotation_component3, rotation_component4, pres_EFIT01_component1, pres_EFIT01_component2, pinj, tinj, ipsiptargt, bt_magnitude, bt_is_positive, gasA, aminor_EFIT01, tritop_EFIT01, tribot_EFIT01, kappa_EFIT01, rmaxis_EFIT01, zmaxis_EFIT01, ech_pwr_total

## E  BASELINES FOR EXPERIMENTAL SECTION

**Calibration Loss (Cali):** Calibration loss Chung et al. (2021) optimizes for an empirical calibration objective,

$$\mathcal{C}(D, \hat{\mathbb{Q}}_p, p) = \mathbb{I}\{\hat{p}_{avg}^{\mathrm{obs}} < p\} * \frac{1}{N} \sum_{i=1}^{N} [(y_i - \hat{\mathbb{Q}}_p(x_i))\mathbb{I}\{y_i > \hat{\mathbb{Q}}_p(x_i)\}]$$

where $\hat{\mathbb{Q}}_p$ is the quantile prediction for quantile level $p \in (0, 1)$, and $\hat{p}_{avg}^{\mathrm{obs}}$ is the average portion of observed data that lies below the quantile prediction, $\hat{p}_{avg}^{\mathrm{obs}} = \frac{1}{N} \sum_{i=1}^{N} [y_i \leq \hat{\mathbb{Q}}_p(x_i)]$. This loss is minimized by an average calibration solution on $D$. During training, the model is trained on uniformly random probabilities, i.e., the training objective is $\mathbb{E}_{p \sim \mathrm{Unif}(0,1)} \mathcal{C}(D, \hat{\mathbb{Q}}_p, p)$.

**Interval Score (Interval):** Interval score Chung et al. (2021) incentivizes the model to output a centered prediction interval (PI) by minimizing the loss,

$$S_\alpha(\hat{l}, \hat{u}; y) = (\hat{u} - \hat{l}) + \frac{2}{\alpha}(\hat{l} - y)\mathbb{I}\{y < \hat{l}\} + \frac{2}{\alpha}(y - \hat{u})\mathbb{I}\{y > \hat{u}\}$$

where $(\hat{l} = \hat{\mathbb{Q}}_{\frac{\alpha}{2}}(x), \hat{u} = \hat{\mathbb{Q}}_{1-\frac{\alpha}{2}}(x))$ is the $(1 - \alpha)$ centered PI. To train for all quantile levels, the training objective is $\mathbb{E}_{\alpha \sim \mathrm{Unif}(0,1)} S_\alpha$.

**Model Agnostic Quantile Regression (MAQR):** MAQR Chung et al. (2021) takes as input the training data along with a trained regression model $\hat{f}(x)$ which is used to construct a dataset of residuals. By assuming nearby points in $\mathcal{X}$ will have similar conditional distributions, an empirical CDF is constructed to model the distribution of residuals given $x$. From this empirical CDF, a quantile model $\hat{g}(x, p)$ is trained, yielding a final quantile prediction $\hat{\mathbb{Q}}_p(x) = \hat{f}(x) + \hat{g}(x, p)$.

**CaliPSo:** With CaliPSo Capone et al. (2025), the quantile model is optimized exclusively for sharpness while maintaining marginal calibration at all times during training. Quantile models are trained on a subset of data $D_{2\delta}$ selected for a centered prediction interval, $(\delta, 1 - \delta)$:

$$D_{2\delta} = \begin{cases} \{(x_i, y_i) \in D | \hat{q}_\delta^{\mathrm{cal}}(x_i) \leq y_i \leq q_{1-\delta}^{\mathrm{cal}}(x_i)\}, & 0 < \delta < 0.5 \\ D, & \delta = 0 \end{cases}$$

where $\hat{q}_p^{\mathrm{cal}} = \hat{q}_0(x) + \alpha_p(\hat{q}_1(x) - \hat{q}_0(x))$, and $\alpha_p = \mathrm{quantile}(p, \frac{y - \hat{q}_0(x)}{\hat{q}_1(x) - \hat{q}_0(x)})$. Model predictions are shifted such that they lie above or below $D_{2\delta}$:

$$\hat{q}_\delta(x) = \min_{x_i, y_i \in D_{2\delta}} (y_i - \hat{q}_\delta^{\mathrm{uncal}}(x_i)) + \hat{q}_\delta^{\mathrm{uncal}}(x), \quad \hat{q}_{1-\delta}(x) = \max_{x_i, y_i \in D_{2\delta}} (y_i - \hat{q}_{1-\delta}^{\mathrm{uncal}}(x_i)) + \hat{q}_{1-\delta}^{\mathrm{uncal}}(x)$$

Models are trained to minimize average error, $\mathbb{E}_{x_i, y_i \sim D_{2\delta}} (||\hat{q}_\delta(x_i) - y_i||_1)$.

**Quantile Recalibration Training (QRT):** QRT Dheur & Ben taieb (2024) trains a model to predict the CDF $F_\theta(Y|X)$. During training, negative log-likelihood is used to optimize the differentiable recalibration map $\Phi_\theta^{\mathrm{REFL}}$ composed with $F_\theta$, which is implemented with the loss:

$$\mathcal{L}(\theta) = -\frac{1}{B} \sum_{i=1}^{B} \log f_\theta(Y_i|X_i) + \alpha \log \phi_\theta^{\mathrm{REFL}}(F_\theta(Y_i|X_i))$$

for a batch of size $B$. For our tests, we use hyperparameters $\alpha = 1$ and $C = False$, meaning the trained CDF model $F_\theta$ is the training output.

## F PLOTS FOR RECALIBRATED MODELS

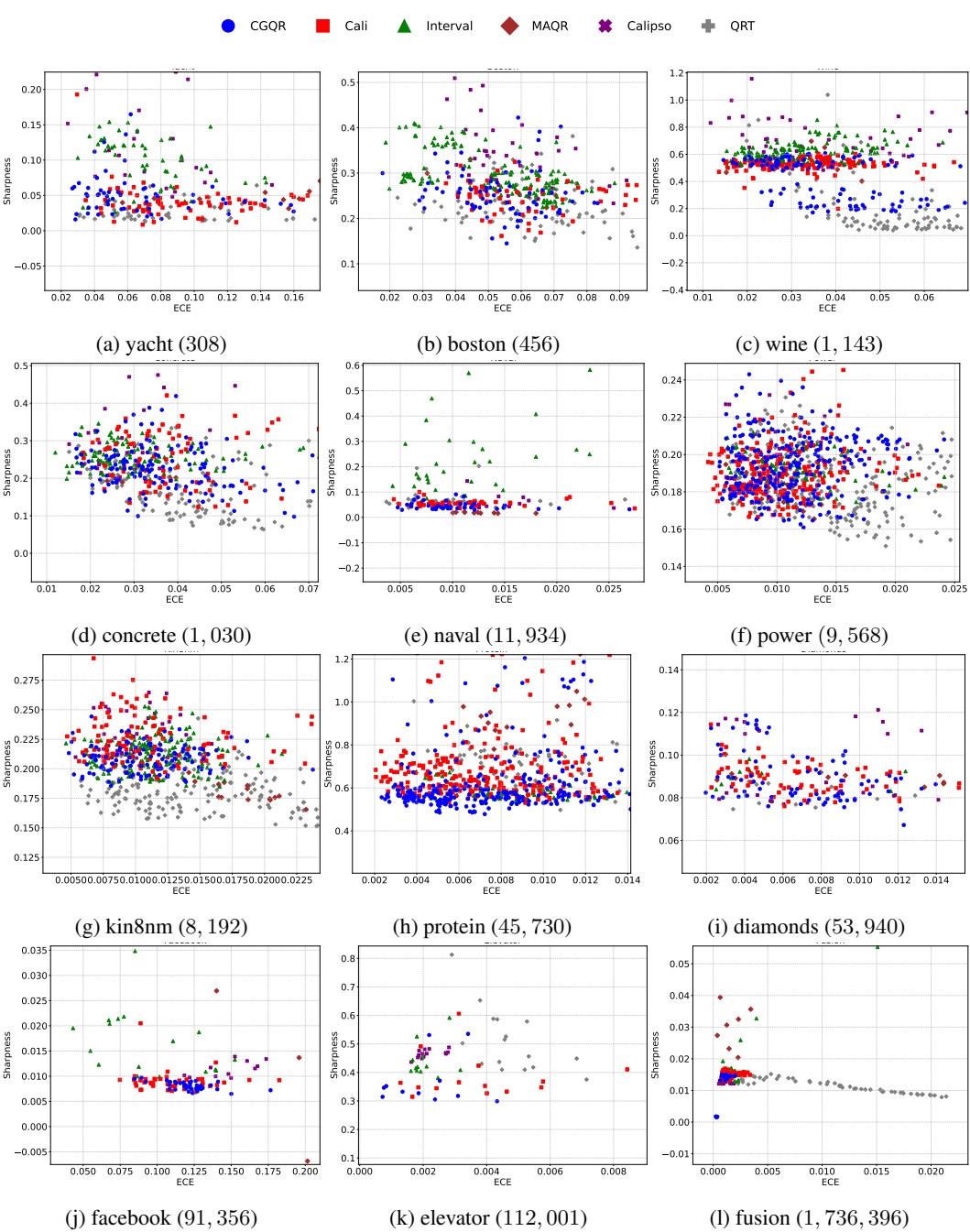

Figure 4: Test time performance of methods in terms of ECE (x-axis) vs Sharpness (y-axis) for different datasets and methods after recalibration. The number in Lower is better for both metrics. The numbers in brackets indicate the dataset size.

## G FULL PARETO METRICS

Table 2: Pareto front metrics in calibration and sharpness space. We report the IGD, GD, and normalized HV, as described in Subsection 2.4. Lower is better for GD and IGD, higher is better for HV. Our method (CGQR) is evaluated, along with Cali, Interval and Model Agnostic Quantile Regression (MAQR) Chung et al. (2021), CaliPSo Capone et al. (2025), and QRT Dheur & Ben taieb (2024). Lower is better for all metrics. The best results across methods are highlighted in bold.

| | | CGQR | CALI | INTERVAL | MAQR | CALIPSO | QRT |
|---|---|---|---|---|---|---|---|
| WINE | IGD ($\downarrow$) | **1.09 ± 0.14** | 2.08 ± 0.19 | 1.83 ± 0.29 | 5.30 ± 0.57 | 2.93 ± 0.50 | **1.04 ± 0.13** |
| | GD ($\downarrow$) | **0.67 ± 0.05** | 1.27 ± 0.11 | **0.72 ± 0.04** | 1.23 ± 0.12 | 0.98 ± 0.19 | **0.73 ± 0.13** |
| | HV ($\uparrow$) | 0.86 ± 0.02 | 0.76 ± 0.02 | 0.74 ± 0.05 | 0.38 ± 0.03 | 0.60 ± 0.02 | **0.90 ± 0.01** |
| CONCRETE | IGD ($\downarrow$) | **0.31 ± 0.04** | 0.83 ± 0.10 | 1.54 ± 0.41 | 1.74 ± 0.23 | 1.35 ± 0.16 | **0.37 ± 0.06** |
| | GD ($\downarrow$) | **0.25 ± 0.03** | 0.48 ± 0.08 | **0.48 ± 0.25** | 0.53 ± 0.18 | 0.44 ± 0.01 | 0.43 ± 0.08 |
| | HV ($\uparrow$) | **0.95 ± 0.00** | 0.76 ± 0.01 | 0.74 ± 0.06 | 0.68 ± 0.02 | 0.76 ± 0.04 | 0.90 ± 0.02 |
| POWER | IGD ($\downarrow$) | **0.17 ± 0.05** | 0.31 ± 0.07 | 1.35 ± 0.21 | 1.69 ± 0.20 | 1.93 ± 0.06 | 0.24 ± 0.01 |
| | GD ($\downarrow$) | 0.05 ± 0.01 | **0.03 ± 0.01** | 0.26 ± 0.05 | 0.29 ± 0.17 | 0.92 ± 0.31 | 0.15 ± 0.01 |
| | HV ($\uparrow$) | **0.97 ± 0.00** | 0.91 ± 0.02 | 0.59 ± 0.09 | 0.49 ± 0.02 | 0.39 ± 0.04 | 0.88 ± 0.01 |
| ENERGY | IGD ($\downarrow$) | **0.24 ± 0.05** | 0.63 ± 0.04 | 0.88 ± 0.10 | 1.65 ± 0.25 | 0.99 ± 0.04 | **0.24 ± 0.06** |
| | GD ($\downarrow$) | **0.18 ± 0.06** | 0.31 ± 0.06 | 0.66 ± 0.14 | **0.22 ± 0.06** | 0.94 ± 0.30 | **0.13 ± 0.03** |
| | HV ($\uparrow$) | 0.94 ± 0.01 | 0.75 ± 0.02 | 0.80 ± 0.04 | 0.44 ± 0.08 | 0.68 ± 0.06 | **0.97 ± 0.01** |
| KIN8NM | IGD ($\downarrow$) | **0.31 ± 0.08** | **0.44 ± 0.07** | **0.49 ± 0.11** | 1.88 ± 0.14 | 1.70 ± 0.10 | **0.38 ± 0.01** |
| | GD ($\downarrow$) | 0.05 ± 0.00 | 0.13 ± 0.03 | 0.22 ± 0.03 | 0.88 ± 0.06 | 0.45 ± 0.20 | 0.35 ± 0.03 |
| | HV ($\uparrow$) | **0.93 ± 0.01** | 0.81 ± 0.02 | 0.85 ± 0.03 | 0.48 ± 0.02 | 0.67 ± 0.02 | 0.81 ± 0.01 |
| DIAMONDS | IGD ($\downarrow$) | **0.31 ± 0.03** | 0.42 ± 0.01 | 0.87 ± 0.05 | 0.85 ± 0.05 | 1.07 ± 0.03 | 0.82 ± 0.03 |
| | GD ($\downarrow$) | 0.03 ± 0.00 | **0.02 ± 0.01** | 0.58 ± 0.11 | 0.66 ± 0.10 | 0.19 ± 0.06 | 1.63 ± 0.34 |
| | HV ($\uparrow$) | **0.95 ± 0.01** | 0.84 ± 0.02 | 0.57 ± 0.05 | 0.59 ± 0.01 | 0.65 ± 0.01 | 0.47 ± 0.04 |
| NAVAL | IGD ($\downarrow$) | 5.76 ± 2.66 | 5.81 ± 2.65 | 4.92 ± 0.93 | **1.36 ± 0.42** | **1.11 ± 0.18** | 5.69 ± 2.08 |
| | GD ($\downarrow$) | 0.87 ± 0.29 | 1.10 ± 0.36 | 8.79 ± 2.50 | **0.01 ± 0.01** | 0.42 ± 0.14 | 1.98 ± 0.17 |
| | HV ($\uparrow$) | **0.96 ± 0.01** | **0.94 ± 0.01** | 0.75 ± 0.09 | **0.96 ± 0.02** | **0.96 ± 0.01** | 0.87 ± 0.02 |
| ELEVATOR | IGD ($\downarrow$) | **0.70 ± 0.13** | **0.66 ± 0.23** | 4.37 ± 0.35 | 3.16 ± 0.03 | 5.51 ± 0.25 | 7.59 ± 0.14 |
| | GD ($\downarrow$) | **0.45 ± 0.07** | **0.84 ± 0.58** | 5.03 ± 0.08 | 5.71 ± 1.54 | 4.83 ± 0.12 | 12.22 ± 0.15 |
| | HV ($\uparrow$) | 0.95 ± 0.02 | **0.99 ± 0.00** | 0.71 ± 0.12 | 0.27 ± 0.04 | 0.70 ± 0.11 | 0.29 ± 0.07 |
| YACHT | IGD ($\downarrow$) | **0.84 ± 0.24** | **0.66 ± 0.15** | 1.23 ± 0.30 | 2.60 ± 0.84 | 0.97 ± 0.08 | **0.74 ± 0.12** |
| | GD ($\downarrow$) | **0.71 ± 0.24** | **0.75 ± 0.25** | **0.86 ± 0.10** | 1.75 ± 0.79 | 0.99 ± 0.11 | **0.62 ± 0.19** |
| | HV ($\uparrow$) | **0.91 ± 0.07** | **0.89 ± 0.02** | 0.74 ± 0.05 | 0.58 ± 0.13 | 0.78 ± 0.02 | **0.85 ± 0.04** |
| PROTEIN | IGD ($\downarrow$) | **0.36 ± 0.08** | **0.39 ± 0.09** | 3.92 ± 1.13 | 4.47 ± 1.15 | 5.19 ± 0.03 | 1.80 ± 0.17 |
| | GD ($\downarrow$) | **0.06 ± 0.01** | 0.16 ± 0.03 | 1.17 ± 0.33 | 3.13 ± 0.19 | 6.35 ± 1.06 | 0.35 ± 0.12 |
| | HV ($\uparrow$) | **0.98 ± 0.01** | 0.89 ± 0.01 | 0.54 ± 0.12 | 0.32 ± 0.02 | 0.33 ± 0.07 | 0.78 ± 0.03 |
| FACEBOOK | IGD ($\downarrow$) | **2.68 ± 0.95** | **2.64 ± 0.91** | 3.12 ± 0.92 | **1.74 ± 0.25** | 3.30 ± 1.26 | 10.46 ± 1.99 |
| | GD ($\downarrow$) | **0.04 ± 0.01** | **0.04 ± 0.01** | 0.32 ± 0.10 | 2.05 ± 0.49 | 0.35 ± 0.07 | 324.78 ± 173.62 |
| | HV ($\uparrow$) | 0.68 ± 0.06 | 0.75 ± 0.06 | **0.92 ± 0.06** | 0.56 ± 0.10 | 0.74 ± 0.08 | 0.56 ± 0.19 |
| BOSTON | IGD ($\downarrow$) | 0.77 ± 0.13 | 1.52 ± 0.18 | **0.53 ± 0.07** | 3.23 ± 0.39 | 1.32 ± 0.08 | 0.83 ± 0.11 |
| | GD ($\downarrow$) | **0.22 ± 0.03** | 0.29 ± 0.06 | **0.27 ± 0.04** | 2.38 ± 0.72 | 0.77 ± 0.12 | 0.93 ± 0.12 |
| | HV ($\uparrow$) | **0.88 ± 0.01** | 0.68 ± 0.04 | **0.89 ± 0.01** | 0.39 ± 0.08 | 0.70 ± 0.02 | 0.78 ± 0.04 |
| FUSION | IGD ($\downarrow$) | **0.17 ± 0.05** | 0.48 ± 0.22 | 0.46 ± 0.08 | 2.12 ± 0.86 | 0.86 ± 0.38 | **0.22 ± 0.10** |
| | GD ($\downarrow$) | **0.00 ± 0.00** | 0.95 ± 0.66 | 0.57 ± 0.22 | 2.15 ± 1.46 | 0.07 ± 0.05 | 0.84 ± 0.56 |
| | HV ($\uparrow$) | **0.96 ± 0.03** | 0.82 ± 0.05 | 0.54 ± 0.13 | 0.37 ± 0.00 | 0.64 ± 0.01 | 0.72 ± 0.02 |

## H    PARETO FRONT ANALYSIS FOR OTHER SHARPNESS METRICS

We tested the performance of all methods using two additional sharpness metrics: the mean predictive interval width (MPIW), which measures the average interval length across all centered intervals, and the average standard deviation of the predictive distribution (STD). Our method performs best also according to these metrics:

| | Metric | CGQR | Cali | Interval | MAQR | Calipso | QRT |
|---|---|---|---|---|---|---|---|
| MPIW | IGD ($\downarrow$) | **0.047 $\pm$ 0.0078** | 0.067 $\pm$ 0.0093 | 0.075 $\pm$ 0.0046 | 0.099 $\pm$ 0.0091 | 0.12 $\pm$ 0.0081 | 0.071 $\pm$ 0.0080 |
| | GD ($\downarrow$) | **0.014 $\pm$ 0.0014** | 0.022 $\pm$ 0.0031 | 0.064 $\pm$ 0.0079 | 0.026 $\pm$ 0.0041 | 0.055 $\pm$ 0.0064 | 0.068 $\pm$ 0.0084 |
| | HV ($\uparrow$) | **0.91 $\pm$ 0.0064** | 0.81 $\pm$ 0.0093 | 0.80 $\pm$ 0.011 | 0.68 $\pm$ 0.0147 | 0.71 $\pm$ 0.0146 | 0.77 $\pm$ 0.016 |

| | Metric | CGQR | Cali | Interval | MAQR | Calipso | QRT |
|---|---|---|---|---|---|---|---|
| STD | IGD ($\downarrow$) | **0.052 $\pm$ 0.0071** | 0.080 $\pm$ 0.0078 | 0.070 $\pm$ 0.0064 | 0.12 $\pm$ 0.011 | 0.15 $\pm$ 0.0129 | 0.063 $\pm$ 0.0075 |
| | GD ($\downarrow$) | **0.024 $\pm$ 0.0028** | 0.032 $\pm$ 0.0038 | 0.059 $\pm$ 0.0083 | 0.034 $\pm$ 0.0039 | 0.038 $\pm$ 0.0043 | 0.090 $\pm$ 0.012 |
| | HV ($\uparrow$) | **0.91 $\pm$ 0.0059** | 0.81 $\pm$ 0.0090 | **0.90 $\pm$ 0.0074** | 0.73 $\pm$ 0.015 | 0.83 $\pm$ 0.0119 | 0.82 $\pm$ 0.013 |

## I    FEASIBILITY ANALYSIS FOR DIFFERENT $\epsilon$

Table 3: Median, upper, and lower deciles of the smallest value of $\epsilon$ for which Algorithm 1 finds a solution. The statistics correspond to 10 seeds.

| Dataset | 10th percentile | median | 90th percentile |
|---|---|---|---|
| boston | 0.0307 | 0.0390 | 0.0624 |
| concrete | 0.0197 | 0.0255 | 0.0296 |
| energy | 0.0203 | 0.0243 | 0.0272 |
| kin8nm | 0.0191 | 0.0224 | 0.0246 |
| naval | 0.00926 | 0.0144 | 0.0181 |
| power | 0.00678 | 0.00842 | 0.0103 |
| protein | 0.00731 | 0.00891 | 0.0115 |
| wine | 0.0228 | 0.0248 | 0.0330 |
| yacht | 0.0208 | 0.0252 | 0.0267 |
| diamonds | 0.00680 | 0.00828 | 0.00901 |
| elevator | 0.0111 | 0.0123 | 0.0133 |
| fusion | 0.0135 | 0.0149 | 0.0171 |
| facebook | 0.0109 | 0.0125 | 0.0145 |

## J SHARPNESS AND ECE DURING TRAINING

Here we visualize how ECE and the average $95\%$ interval length behave during training for three different datasets and two seeds. We consistently observed that ECE decreases before increasing again, with a minimum occurring between 25 and 50 epochs, whereas sharpness decreases throughout training.

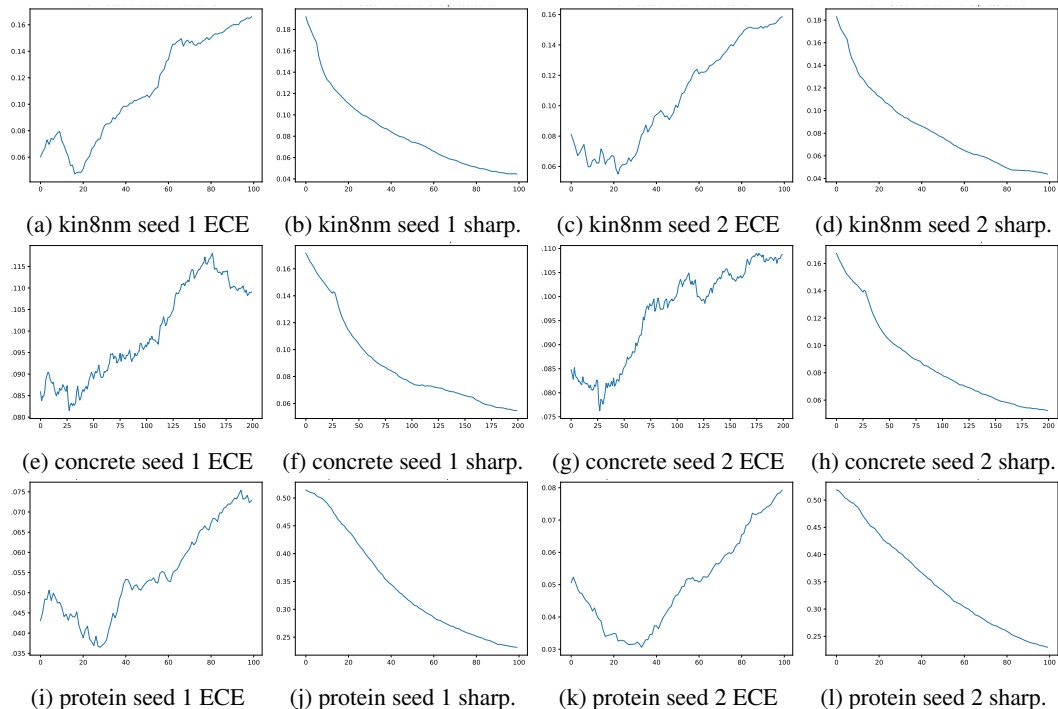

(a) kin8nm seed 1 ECE    (b) kin8nm seed 1 sharp.    (c) kin8nm seed 2 ECE    (d) kin8nm seed 2 sharp.

(e) concrete seed 1 ECE    (f) concrete seed 1 sharp.    (g) concrete seed 2 ECE    (h) concrete seed 2 sharp.

(i) protein seed 1 ECE    (j) protein seed 1 sharp.    (k) protein seed 2 ECE    (l) protein seed 2 sharp.

Figure 5: Behavior of ECE and sharpness (average $95\%$ confidence interval length) during training.

## K PARETO FRONT RESULTS FOR SMALLER MODEL SIZES

Here we report the experimental results using three different model architectures: 1 hidden layer, 32 hidden dimensions; 2 hidden layers, 64 hidden dimensions; 4 hidden layers, 128 hidden dimensions.

Our method still performs best in the IGD and HV metrics. However, it performs slightly worse than best in the GD metric when using relatively small models (up to two layers and 64 hidden dimensions).

### K.1 1 HIDDEN LAYER, 32 HIDDEN DIMENSIONS:

|  | CGQR | Cali | Interval | MAQR | Calipso | QRT |
|---|---|---|---|---|---|---|
| IGD ($\downarrow$) | $\mathbf{0.0384 \pm 0.0092}$ | $0.0423 \pm 0.0095$ | $0.0427 \pm 0.0088$ | $0.152 \pm 0.043$ | $0.189 \pm 0.025$ | $0.156 \pm 0.016$ |
| GD ($\downarrow$) | $0.0188 \pm 0.0037$ | $0.0159 \pm 0.0049$ | $0.0249 \pm 0.0045$ | $\mathbf{0.0102 \pm 0.0032}$ | $0.077 \pm 0.022$ | $0.152 \pm 0.026$ |
| HV ($\uparrow$) | $\mathbf{0.871 \pm 0.020}$ | $0.848 \pm 0.021$ | $0.633 \pm 0.038$ | $0.770 \pm 0.046$ | $0.298 \pm 0.033$ | $0.235 \pm 0.025$ |

### K.2 2 HIDDEN LAYERS, 64 HIDDEN DIMENSIONS:

|  | CGQR | Cali | Interval | MAQR | Calipso | QRT |
|---|---|---|---|---|---|---|
| IGD ($\downarrow$) | $\mathbf{0.073 \pm 0.012}$ | $0.083 \pm 0.014$ | $0.13 \pm 0.014$ | $0.14 \pm 0.022$ | $0.17 \pm 0.020$ | $0.13 \pm 0.0081$ |
| GD ($\downarrow$) | $0.032 \pm 0.0075$ | $\mathbf{0.029 \pm 0.0051}$ | $0.13 \pm 0.016$ | $0.072 \pm 0.029$ | $0.088 \pm 0.022$ | $0.098 \pm 0.011$ |
| HV ($\uparrow$) | $\mathbf{0.91 \pm 0.011}$ | $0.85 \pm 0.018$ | $0.65 \pm 0.023$ | $0.65 \pm 0.037$ | $0.66 \pm 0.047$ | $0.59 \pm 0.038$ |

## K.3  4 HIDDEN LAYERS, 128 HIDDEN DIMENSIONS:

|  | CGQR | Cali | Interval | MAQR | Calipso | QRT |
|---|---|---|---|---|---|---|
| IGD (↓) | **0.036 ± 0.0051** | 0.054 ± 0.0059 | 0.12 ± 0.0094 | 0.17 ± 0.016 | 0.13 ± 0.013 | 0.066 ± 0.0091 |
| GD (↓) | **0.021 ± 0.0027** | 0.030 ± 0.0043 | 0.085 ± 0.013 | 0.10 ± 0.016 | 0.068 ± 0.010 | 0.067 ± 0.011 |
| HV (↑) | **0.93 ± 0.0069** | 0.83 ± 0.015 | 0.74 ± 0.023 | 0.56 ± 0.040 | 0.69 ± 0.028 | 0.80 ± 0.025 |

