# OpenReview forum: "Calibration-Guided Quantile Regression"
_ICLR.cc/2026/Conference — Submitted to ICLR 2026_

### Official Review · Reviewer_t1Nn · 2025-11-01

**Soundness:** 1
**Presentation:** 2
**Contribution:** 1
**Rating:** 0
**Confidence:** 4

**Summary:**

The authors emphasize the importance of calibration in quantile regression and propose to minimize the quantile loss subject to the constraint that the calibration error on a validation subset remains below a specified threshold. However, the algorithm and accompanying theory do not appear to actually solve this constrained optimization problem, and the procedure as written provides no guarantee that it will converge to a solution that satisfies the constraint. The authors also claim that their method performs fitting and calibration simultaneously and may therefore improve upon two-stage approaches that first train a model and then apply post-hoc calibration. As presented, however, this claim is not convincingly supported by either the algorithmic design or the theoretical results.

**Strengths:**

The authors recognize that calibration is important in quantile regression and propose an objective that is to minimize the quantile loss under the constraint that the solution is calibrated up to some threshold.

**Weaknesses:**

1. The claim and key motivation of the paper that post-hoc calibration necessarily worsens `"sharpness'' or "discrimination'' is not well justified. Standard calibration methods—such as Platt scaling in classification—fit a secondary model that minimizes a proper loss (e.g., log-likelihood) on validation data, and therefore aim to improve predictive quality rather than degrade it. In the context of quantile regression, an analogous calibration step could be performed using the quantile loss itself. While calibration may increase interval width when the original model is overconfident, this is a correction for miscalibration rather than a flaw of post-hoc calibration methods.



2. In Algorithm~1, it is unclear why the procedure would ever converge to a valid solution satisfying the constraint of objective (12). The gradient descent update to the parameter $\theta^\*$ occurs only when the validation ECE falls below $\varepsilon$  and the sharpness decreases, but there is no guarantee that this condition will ever be met. If it is not, the algorithm simply returns the initial parameter $\theta$, regardless of how many gradient steps are taken. For small values of $\varepsilon$, there is no reason to expect gradient descent to produce a model that satisfies both constraints, and the algorithm provides no theoretical or practical mechanism ensuring feasibility, termination, or even monotonic improvement.

3. The proposed objective in (12) seeks to minimize the loss subject to the constraint that the calibration error is at most $\varepsilon$. However, Algorithm~1 does not actually optimize this constrained objective. Instead, it performs unconstrained gradient descent on the loss and then selects, from what may be an empty set, the iterates that happen to satisfy the calibration constraint. This is not equivalent to solving the constrained optimization problem in (12). A principled approach would require directly minimizing the loss over the feasible region---that is, performing a true constrained optimization in which the calibration constraint is enforced throughout the procedure. Although this seems to be the conceptual goal of the paper, it is neither implemented nor addressed algorithmically, and carrying it out in practice is nontrivial.

4. Theorem 3.2 does not appear to be useful in its current form, and it may even be incorrect. It is unclear whether $m(\varepsilon)$ is intended to denote the number of gradient descent steps required for convergence, but as noted above, the procedure may not converge at all, in which case the theorem would be false as stated. Even if convergence were guaranteed, $m(\varepsilon)$ is likely to be extremely large in practice, making the stated high-probability guarantee effectively vacuous: the error term $\gamma$ would need to be so large for the bound to hold with high probability that the result provides no meaningful assurance about the returned model. Substantially stronger theoretical development is needed. Algorithm~1 is too generic to support the type of guarantee claimed in Theorem 3.2. A useful theory would need to answer basic questions such as: under what conditions does the method converge, and at what rate? In general, no such convergence can be expected without additional assumptions, and any rigorous result will likely require specifying a concrete optimization procedure (e.g., gradient descent with fixed step size on a convex objective) and proving guarantees in that setting.

**Questions:**

My questions are in direct reference to the weaknesses discussed above:

1. Can the authors justify the claim that post-hoc calibration necessarily harms sharpness or discrimination? Are there empirical or theoretical results demonstrating this effect specifically in quantile regression?

2. What guarantees, if any, ensure that Algorithm~1 will produce a parameter satisfying the calibration constraint rather than simply returning the initialization? In practice, does the early-stopping condition ever trigger?

3. How is the constrained optimization problem in (12) being solved, given that Algorithm~1 performs only unconstrained gradient descent and then filters iterates after the fact?

4. In Theorem~3.2, what is the precise meaning of $m(\varepsilon)$, and under what assumptions is convergence to such a point guaranteed?

---

> ### Author Response · Authors · 2025-11-25
>
> Thank you for your thorough review. We have addressed your main comments as follows:
>
> **Claim that Post-Hoc Calibration Harms Sharpness**
>
> We applied post-hoc calibration to the models after being trained to completion with the vanilla quantile loss. We observed that post-hoc calibration consistently and significantly increases sharpness for all except the elevator dataset, where few (less than half) cases yield a marginal improvement in sharpness. Below we report the relative difference in sharpness before and after calibration (i.e., (recal_test_sharpness - test_sharpness) / test_sharpness):
>
> | Dataset   | Median | 10th percentile   | 90th percentile  |
> |-----------|--------|-------|-------|
> | boston    | 3.4    | 2.6   | 3.9   |
> | concrete  | 4.9    | 4.1   | 5.8   |
> | energy    | 4.3    | 3.1   | 5.7   |
> | kin8nm    | 3.3    | 3.2   | 3.7   |
> | naval     | 0.30   | 0.30  | 0.30  |
> | power     | 3.5    | 3.1   | 4.0   |
> | protein   | 5.9    | 4.2   | 7.0   |
> | wine      | 6.8    | 5.3   | 7.9   |
> | yacht     | 4.3    | 1.8   | 5.8   |
> | diamonds  | 1.9    | 1.7   | 2.1   |
> | elevator  | 0.014  | -0.029| 0.057 |
> | fusion    | 5.2    | 5.1   | 5.4   |
> | facebook  | 1.3    | 1.0   | 1.5   |
>
> We note that similar observations have been reported in Chung et al. (2021), and that achieving a favorable trade-off between sharpness and calibration is a topic frequently addressed in the literature (Zhao et al. 2020; Kuleshov et al., 2022; Dheur et al., 2023).
>
>
>
> **Direct Minimization of Equation (12)**
>
> One of the main takeaways from our experimental results is that finding model that minimizes (12) is very difficult: the two loss functions Cali and Interval, which we compare our method against, correspond to a weighted sum of a calibration and sharpness terms. This in turn is very similar to optimizing (12) using a classical Lagrangian reformulation approach. Similarly, QRT enforces calibration on the observed data throughout training, while minimizing the NLL. However, we find that these methods perform poorly due to the presence of local minima, as illustrated by the Pareto front metrics. By contrast, the simultaneous pinball loss, though employing a formulation that is seemingly unrelated to (12), results in the best performance, indicating that it best navigates the loss landscape.
>
> **Satisfaction of Calibration Constraint**
>
> We analyzed the lowest value of $\epsilon$ required to obtain feasibility. Below you can find the results over ten seeds
> | Dataset   | 10th percentile      | Median      | 90th Percentile|
> |-----------|----------|----------|----------|
> | boston    | 0.031   | 0.039   | 0.062   |
> | concrete  | 0.02   | 0.026   | 0.03   |
> | energy    | 0.02   | 0.024   | 0.027   |
> | kin8nm    | 0.019   | 0.022   | 0.025   |
> | naval     | 0.0093  | 0.014   | 0.018   |
> | power     | 0.0068  | 0.0084  | 0.01   |
> | protein   | 0.0073  | 0.0089  | 0.012   |
> | wine      | 0.023   | 0.025   | 0.033   |
> | yacht     | 0.021   | 0.025   | 0.0267   |
> | diamonds  | 0.0068  | 0.0083  | 0.009  |
> | elevator  | 0.011   | 0.012   | 0.013   |
> | fusion    | 0.014   | 0.015   | 0.017   |
> | facebook  | 0.011   | 0.013   | 0.015   |
>
> With the exception of the Boston dataset, which is fairly small, we found that for any value of $\epsilon>0.04$ our algorithm will yield a feasible and valid result with high probability. We have included this discussion in the revised manuscript.
>
> **Meaning of $m(\epsilon)$ and Theorem 3.2**
>
> The symbol $m(\epsilon)$ denotes the number of parameters encountered during training that satisfy the constraints of (12). In order for Theorem 3.2 to hold, we require $m(\epsilon)$ to be larger than 0. In practical terms, this means that $\epsilon$ needs to be sufficiently large, and the loss function must yield a model that renders (12) feasible. However, as discussed above, this frequently holds for $\epsilon>0.04$. Please note that this does not render Theorem 3.2 incompatible with our algorithm.
>
> We agree that our bounds are less tight than those encountered in the calibration literature and the probability that it holds can, in theory, become arbitrarily small. However, Theorem 3.2 also shows that a high $m(\epsilon)$ can countered by increasing the data size. Moreover, our empirical results strongly suggest that the resulting bound is not loose in practice.
>
>
> Zhao, Shengjia, Tengyu Ma, and Stefano Ermon. "Individual calibration with randomized forecasting." ICML 2020.
>
> Chung, Youngseog, Willie Neiswanger, Ian Char, and Jeff Schneider. "Beyond pinball loss: Quantile methods for calibrated uncertainty quantification." Neurips 2021
>
> Kuleshov, Volodymyr, and Shachi Deshpande. "Calibrated and sharp uncertainties in deep learning via density estimation." In ICML 2022.
>
> Dheur, Victor, and Souhaib Ben Taieb. "A large-scale study of probabilistic calibration in neural network regression." ICML 2023.

---

### Official Review · Reviewer_ALqY · 2025-11-01

**Soundness:** 3
**Presentation:** 2
**Contribution:** 3
**Rating:** 6
**Confidence:** 3

**Summary:**

This paper presents CGQR, addressing the fundamental trade-off between calibration and sharpness. The core insight is: instead of the standard train and calibrate approach, which hinders sharpness, CGQR introduces a regularizer lambda*L_calib and an early stopping criterion to trade off between calibration and sharpness by allowing miscalibration up to some pre-specified tolerance. After training, the model is frozen, and a recalibration step is applied. The authors use the Pareto frontier to evaluate. The claim is that CGQR produces better sharpness-calibration trade-offs.

**Strengths:**

1. Simple solution to trace a pareto frontier. Based on my understanding, the method uses lambda as a knob to control how quickly the model learns calibration, which in turn implicitly trades off sharpness by determining the early-stopping point. This is a novel and powerful method for tracing a Pareto frontier.
2. Strong and extensive empirical results, albeit on small UCI datasets.

**Weaknesses:**

1. The methodology needs to be clearer. For example, calibration on the validation dataset and then doing recalibration with a separate calibration set was not immediately clear and was frustrating to navigate. The presentation of the main method should be cleaner and not use the same terminology for separate procedures.
2. Based on my understanding, the entire method hinges on a key assumption that regularizing the loss function with L_calib on D_train will improve performance. This may work on small UCI datasets, but potentially in larger and more multi-dimensional learning, a model could overfit without that learning transferring to D_val. A more comprehensive discussion and analysis of how model overfitting would affect the performance would significantly improve the quality of the work.
3. One of the keys to the method is this lambda*L_calib term as a regularizer. The paper seems to be missing an ablation study comparing the existing method to lambda=0 with the early stopping rule. This ablation would be crucial for isolating the true benefit of your lambda*L_calib term. It may be possible that the D_val stopping rule by itself provides most of the benefit.

**Questions:**

1. Based on weakness 2, your method relies on L_calib on D_train being a better proxy for the true calibration on D_val. Have you ever observed this proxy fail? Is it possible for a model to overfit and minimize L_calib on D_train without this improvement ever translating to D_val, thus causing the early-stopping rule to never be met?
2. In Table 1, IGD for CGQR is 0.03±0.04. Is this a typo? The IGD appears to be quite small compared to other baselines. Furthermore, why is Cali in the IGD row bolded?

---

> ### Author Response · Authors · 2025-11-25
>
> Thank you kindly for your review. We address your comments as follows:
>
> **Terminology**
>
> We have improved the writing and now state that the set $D_{val}$ is used exclusively to assess calibration and sharpness during training, without recalibration.
>
> **IGD in Table 1**
>
> Thank you for pointing this out. This was a typo and has been fixed.
>
> **Generalization to test set**
>
> Theorem 3.2 indicates how we can expect the model to generalize as a function of the number of data and the number of models $m(\epsilon)$ that satisfy the constraints during training. If $m(\epsilon)$ is large, the best model is chosen from a large pool of models, yielding good sharpness but potentially worse ECE on the test set. This can be countered by having a large data set, which tightens the bound and increases the probability of satisfaction. It should be noted, however, that the exchangeability assumption (e.g., iid data) must hold, which is typical for calibration results, but can be a rigid assumption and does not hold in general. An example would be an RL setup.
>
> **Comparison with Vanilla Pinball Training**
>
> We additionally examined the experimental results when using a vanilla training setup for the simultaneous pinball loss (no early stopping). The results show that ECE suffers significantly, while sharpness behaves very well. This roughly corresponds to points on the right hand-side of the Pareto front:
>
> ECE, vanilla pinball loss training:
> | Dataset   | Median | 90th Percentile | 10th Percentile  |
> |-----------|--------|------|------|
> | boston    | 0.10   | 0.12 | 0.046 |
> | concrete  | 0.12   | 0.13 | 0.069 |
> | energy    | 0.11   | 0.14 | 0.076 |
> | kin8nm    | 0.044  | 0.066 | 0.037 |
> | naval     | 0.056  | 0.096 | 0.046 |
> | power     | 0.049  | 0.061 | 0.036 |
> | protein   | 0.029  | 0.038 | 0.024 |
> | wine      | 0.055  | 0.085 | 0.041 |
> | yacht     | 0.097  | 0.15  | 0.062 |
> | diamonds  | 0.017  | 0.025 | 0.010 |
> | elevator  | 0.057  | 0.11  | 0.025 |
> | fusion    | 0.023  | 0.033 | 0.016 |
> | facebook  | 0.10   | 0.14  | 0.082 |
>
>
> Sharpness, vanilla pinball loss training:
> | Dataset   | Median | 90th percentile  | 10th percentile |
> |-----------|--------|------|------|
> | boston    | 0.10   | 0.12 | 0.065 |
> | concrete  | 0.059  | 0.066 | 0.048 |
> | energy    | 0.016  | 0.017 | 0.012 |
> | kin8nm    | 0.11   | 0.12 | 0.10  |
> | naval     | 0.030  | 0.037 | 0.027 |
> | power     | 0.11   | 0.12 | 0.097 |
> | protein   | 0.34   | 0.37 | 0.32  |
> | wine      | 0.27   | 0.28 | 0.22  |
> | yacht     | 0.022  | 0.029 | 0.011 |
> | diamonds  | 0.063  | 0.065 | 0.060 |
> | elevator  | 0.32   | 0.33 | 0.31  |
> | fusion    | 0.0076 | 0.0079 | 0.0073 |
> | facebook  | 0.0058 | 0.0063 | 0.0054 |

---

### Official Review · Reviewer_WBvh · 2025-11-02

**Soundness:** 3
**Presentation:** 2
**Contribution:** 2
**Rating:** 2
**Confidence:** 3

**Summary:**

The paper proposes Calibration-Guided Quantile Regression (CGQR), a simple procedure that explicitly trades-off between calibration and sharpness. The procedure is conducted by (1) minimizing the simultaneous pinball loss during training and (2) on a held-out validation set, keeping the sharpest checkpoint whose empirical ECE is below a specified tolerance. A standard finite-sample analysis is provided. The paper evaluates models by computing Pareto fronts in the calibration-sharpness trade-off(ECE vs. average 95% interval length) and reports hypervolume (HV), GD, and IGD metrics across real datasets. Empirically, CGQR outperforms other methods within Pareto-frontier-related metrics.

**Strengths:**

1. The simultaneous pinball loss is simple and easy to implement.

2. Theoretical guarantee is provided.

3. Empirical results show the effectiveness of reaching Pareto frontiers and obtaining good trade-offs.

**Weaknesses:**

1. The Pareto optimality of this method is the result of always keeping the sharpest checkpoint with respect to 95\% coverage interval length. But if any other sharpness measurement (e.g., 75\% coverage interval length) is to be considered in evaluation, we might not expect the checkpoints are still at the Pareto frontiers or outperforming other methods.

2. The implementation details of how to do gradient descent on the simultaneous pinball loss should not be omitted from the main text. I think that procedure involves an integration with respect to $\tau$ from 0 to 1, which is non-trivial.

3. Theoretical method is standard and not novel.

**Questions:**

1. I don't know if the word "trade-off" best describes the method; the method cannot directly control which level of ECE it will get. Consider we are training the model for infinitely long time. The whole training trajectory is fixed no matter how you select $\epsilon$. The role of $\epsilon$ is just setting the filtering threshold of checkpoints, and the improvement highly relies on the spread/randomness of checkpoints. I would like to see the overall trend along the training trajectories, which I think is the critical property of this method.

2. The final implementation includes three ingredients: (1) the loss, (2) the optimizer, and (3) the model structure. The authors spend almost all the efforts in explaining the first one. But what if you change the optimizer (so that the training trajectory is changed)? Why the optimizer matters: Imagine there is a strong enough oracle optimizer that can reach the minimum of the loss in just one step, then your method will definitely fail in that sense (only one checkpoint now). What if you change the model structure? I think more discussions should be included.

3. Typo: $f(q(X, \tau)+|X)$ in equation (8); Table 1 CALI IGD should not be bold.

---

> ### Author Response · Authors · 2025-11-23
>
> Thank you for your review. We have addressed your comments as follows:
>
>
>
>  **Loss and sharpness along training trajectories**
>
> We visualized the validation ECE and Sharpness during training on different datasets (Appendix J in the revised manuscript). Across all datasets and seeds, ECE first decreases, then increases, whereas sharpness improves throughout training. We also observed that the minimum ECE is typically achieved between 25 and 50 epochs. This is also approximately when our algorithm selects the best model. This strongly suggests that longer training would not significantly change the outcome of our method.
>
>
>
> **Dependence of Pareto performance on metrics**
>
> We tested the performance of all methods using two additional sharpness metrics: the mean predictive interval width (MPIW), which measures the average interval length across all centered intervals, and the average standard deviation of the predictive distribution (STD). Our method performs best also according to these metrics:
>
> MPIW:
> || | CGQR | Cali | Interval | MAQR | Calipso | QRT |
> |-|-|-|-|-|-|-|-|
> | | IGD ↓ | **0.047 ± 0.0078** | 0.067 ± 0.0093 | 0.075 ± 0.0046 | 0.099 ± 0.0091 | 0.12 ± 0.0081 | 0.071 ± 0.008 |
> | | GD ↓ | **0.014 ± 0.0014** | 0.022 ± 0.0031 | 0.064 ± 0.0079 | 0.026 ± 0.0041 | 0.055 ± 0.0064 | 0.068 ± 0.0084 |
> | | HV ↑ | **0.91 ± 0.0064** | 0.81 ± 0.0093 | 0.80 ± 0.011 | 0.68 ± 0.0147 | 0.71 ± 0.0146 | 0.77 ± 0.016 |
>
>
> STD:
> | | | CGQR | Cali | Interval | MAQR | Calipso | QRT |
> |-|-|-|-|-|-|-|-|
> | | IGD ↓ | **0.052 ± 0.0071** | 0.080 ± 0.0078 | 0.070 ± 0.0064 | 0.12 ± 0.011 | 0.15 ± 0.0129 | 0.063 ± 0.0075 |
> | | GD ↓ | **0.024 ± 0.0028** | 0.032 ± 0.0038 | 0.059 ± 0.0083 | 0.034 ± 0.0039 | 0.038 ± 0.0043 | 0.090 ± 0.012 |
> | | HV ↑ | **0.91 ± 0.0059** | 0.81 ± 0.0090 | **0.90 ± 0.0074** | 0.73 ± 0.015 | 0.83 ± 0.0119 | 0.82 ± 0.013 |
>
>
>
> **Gradient Descent on Simultaneous Pinball Loss**
>
> To perform gradient descent on the simultaneous pinball loss, we sample a batch of quantile levels from the uniform distribution between 0 and 1, then perform gradient descent on the sum of the resulting pinball losses. We have included this change in the revised manuscript.
>
>
>
> **Dependence of results on optimizer**
>
> We repeated the experiments using three different optimization algorithms. Our method consistently performs best except when using SGD. However, this is likely because SGD is more prone to getting stuck in local minima than other, more common optimizers:
>
> RMSProp:
> | Metric | CGQR | Cali | Interval | MAQR | Calipso | QRT |
> |-|-|-|-|-|-|-|
> | IGD ↓ | **0.033 ± 0.0097** | 0.075 ± 0.019 | 0.13 ± 0.031 | 0.17 ± 0.034 | 0.14 ± 0.027 | 0.063 ± 0.0093 |
> | GD ↓ | **0.022 ± 0.0074** | 0.050 ± 0.021 | 0.057 ± 0.012 | 0.053 ± 0.014 | 0.033 ± 0.011 | 0.075 ± 0.016 |
> | HV ↑ | **0.93 ± 0.019** | 0.81 ± 0.023 | 0.77 ± 0.066 | 0.67 ± 0.069 | 0.78 ± 0.027 | 0.80 ± 0.034 |
>
> AdamW:
> | Metric | CGQR | Cali | Interval | MAQR | Calipso | QRT |
> |-|-|-|-|-|-|-|
> | IGD ↓ | **0.039 ± 0.011** | 0.074 ± 0.017 | 0.15 ± 0.036 | 0.17 ± 0.030 | 0.15 ± 0.028 | 0.070 ± 0.010 |
> | GD ↓ | **0.025 ± 0.010** | 0.053 ± 0.021 | 0.057 ± 0.011 | 0.057 ± 0.014 | 0.041 ± 0.012 | 0.079 ± 0.016 |
> | HV ↑ | **0.94 ± 0.012** | 0.81 ± 0.024 | 0.77 ± 0.065 | 0.67 ± 0.068 | 0.78 ± 0.026 | 0.80 ± 0.034 |
>
> SGD:
> | Metric | CGQR | Cali | Interval | MAQR | Calipso | QRT |
> |-|-|-|-|-|-|-|
> | IGD ↓ | 0.13 ± 0.023 | 0.061 ± 0.016 | 0.097 ± 0.018 | 0.15 ± 0.029 | 0.12 ± 0.029 | **0.057 ± 0.012** |
> | GD ↓ | 0.062 ± 0.0068 | 0.030 ± 0.011 | 0.050 ± 0.016 | 0.051 ± 0.015 | **0.021 ± 0.0068** | 0.063 ± 0.017 | 0.069 ± 0.013 |
> | HV ↑ | 0.71 ± 0.026 | **0.83 ± 0.026** | 0.78 ± 0.066 | 0.68 ± 0.068 | 0.79 ± 0.025 | 0.81 ± 0.034 | 0.82 ± 0.023 |
>
>
> **Dependence of Results  on Model Architecture**:
>
> We additionally reran the experiment using three different model architectures:
> - 1 hidden layer, 32 hidden dimensions
> - 2 hidden layers, 64 hidden dimensions,
> - 4 hidden layers, 128 hidden dimensions
>
> We have included the results in Appendix K (omitted here due to space issues). Our method still performs best across all Pareto front metrics except for the two smaller models, where it is best in IGD and HV, and close to best in GD. We note the results are generally worse across all methods when using the smaller models, as discussed in Appendix C.

---

> > ### Comment · Reviewer_WBvh · 2025-11-26
> >
> > Thank the authors for the additional experiments and information. From my point of view, my concerns are:
> >
> > 1. From the explanations to my W2 (implementation of selecting $\tau$ in the pinball loss), the method adopted is uniformly sampling a batch of $\tau$'s. Yet that approach I think is quite common in quantile regression, and that approach alone (if with sufficiently many data) will be calibrated. In that sense, selecting some ``best'' checkpoint along the training trajectory that meets the Pareto frontier is kind of artifact.
> >
> > 2. From the experiments answering my Q2 (optimizer and model structure), the results are worrying. The success of finding a good checkpoint relies heavily on the optimizer selected and that lacks convincing intuition/understanding behind the dynamics. I agree with Reviewer t1Nn that the existence of a good checkpoint is not guaranteed. In that sense, the method is not concrete enough.
> >
> > Thank the authors again for the efforts.

---

> > > ### Author Response · Authors · 2025-12-04
> > >
> > > Thank you kindly for reviewing our rebuttal. We address your further concerns as follows:
> > >
> > > **Calibration of Simultaneous Quantile Regression and Selection of Best Checkpoint**
> > >
> > > Although the minimizer of the simultaneous pinball loss does exhibit calibration for infinite data, our paper considers the finite-data case, where this is decidedly not the case. This setting requires special attention and is especially relevant because data is always finite in practice. The need to differentiate between both settings is also evidenced by the results obtained when using a vanilla training setup for the simultaneous pinball loss (no early stopping), where ECE suffers significantly (please see the our response to reviewer ALqY titled Comparison with Vanilla Pinball Training). This shortcoming has motivated several works that address calibration with finite data, also using loss functions similar to the pinball loss (Romano et al., 2019; Zhao et al., 2020; Chung et al., 2021). Hence, picking the best point is not an "artifact" of our setup, but rather a practically relevant research question.
> > >
> > >
> > > **Dependence of Results on Optimizer**
> > >
> > > Thank you for this observation. We reran the experiments for SGD using more seeds (10 instead of 3, which we first chose in the interest of time). The results are as follows (please note that the results are normalized):
> > >
> > > | Metric      | CGQR               | Cali               | Interval       | MAQR                 | Calipso        | QRT            |
> > > | ----------- | ------------------ | ------------------ | -------------- | -------------------- | -------------- | -------------- |
> > > | IGD (↓) | **0.022 ± 0.0025** | **0.019 ± 0.0016** | 0.064 ± 0.0064 | **0.022 ± 0.0024**   | 0.093 ± 0.0088 | 0.041 ± 0.0051 |
> > > | GD (↓)  | 0.011 ± 0.0015     | 0.011 ± 0.0020     | 0.11 ± 0.027   | **0.0046 ± 0.00089** | 0.068 ± 0.013  | 0.053 ± 0.013  |
> > > | HV (↑)  | **0.85 ± 0.017**   | 0.78 ± 0.013       | 0.62 ± 0.043   | **0.84 ± 0.022**     | 0.50 ± 0.034   | 0.73 ± 0.037   |
> > >
> > >
> > > Our approach still performs joint best on all metrics. While we agree that this result indicates a sensitivity to the optimizer, our approach still performs best across optimizers. Moreover, we stress that it performs significantly better than other methods when using Adam and AdamW, which are more likely to be used as off-the-shelf optimizers than SGD, since they require less hyperparameter-tuning to obtain good results.
> > >
> > >
> > > Romano, Yaniv, Evan Patterson, and Emmanuel Candes. "Conformalized quantile regression." Advances in neural information processing systems 32 (2019).
> > >
> > > Zhao, Shengjia, Tengyu Ma, and Stefano Ermon. "Individual calibration with randomized forecasting." ICML 2020.
> > >
> > > Chung, Youngseog, Willie Neiswanger, Ian Char, and Jeff Schneider. "Beyond pinball loss: Quantile methods for calibrated uncertainty quantification." Neurips 2021

---

### Official Review · Reviewer_NdLJ · 2025-11-09

**Soundness:** 2
**Presentation:** 3
**Contribution:** 1
**Rating:** 4
**Confidence:** 4

**Summary:**

The paper proposes a method (CGQR) for regression uncertainty quantification that aims to explicitly balance calibration and sharpness. The approach trains a quantile regression model using the pinball loss and selects the sharpest model that satisfies a user specified calibration error threshold (ECE \leq \epsilon), thereby tracing out the Pareto front between calibration and sharpness. The authors provide finite sample guarantees and evaluate their method on several standard regression datasets and a nuclear fusion prediction task.

**Strengths:**

1. The paper is clearly written and well structured.
2. The motivation of balancing calibration and sharpness in predictive uncertainty is relevant for practical applications.
3. The method is simple, easy to implement, and provides explicit control over the calibration sharpness trade-off.
4. The empirical evaluation is thorough, covering a range of datasets and comparing with several baselines.

**Weaknesses:**

1. The main contribution is a straightforward combination of standard quantile regression (pinball loss minimization) with a validation set based selection for calibration. The Pareto front analysis is more diagnostic than methodological. There is no new loss function, model architecture, or significant algorithmic contribution.
2. The theoretical results are direct applications of standard concentration inequalities and union bounds, and do not provide new insights or tight guarantees.
3. The method does not enforce non crossing quantiles, which can be problematic for interpretability. The choice of miscalibration tolerance (\epsilon) is left to the practitioner, with little guidance or analysis of sensitivity.

**Questions:**

1. The paper frames the calibration sharpness trade-off as a constrained optimization problem, but the solution is essentially a validation-set-based selection. Is there a principled way to jointly optimize for both objectives, perhaps via a Lagrangian or multi-objective optimization framework? How does the proposed approach compare, in theory, to such alternatives?
2. The method does not enforce non-crossing quantiles, which can lead to interpretability issues. Are there theoretical guarantees or practical strategies to ensure monotonicity of quantile functions within the CGQR framework? How would enforcing non crossing constraints affect the calibration and sharpness trade off?
3. The method relies on a user specified miscalibration tolerance. Can the authors provide theoretical or empirical guidance on how to select \epsilon? How sensitive are the results to this parameter, and what are the implications for model selection in practice?
4. The approach is built around the pinball loss for quantile regression. How would the theoretical guarantees and Pareto front analysis extend to other uncertainty quantification frameworks? Are there limitations to the generality of the proposed method?
5. How does the finite-sample coverage guarantee of CGQR compare theoretically to conformal prediction methods, which provide distribution-free coverage guarantees? Can the authors clarify the advantages or limitations of their approach in this context?

---

> ### Author Response · Authors · 2025-11-21
>
> Thank you kindly for your review. We have addressed your questions as described below.
>
> **Comparison to Lagrangian and multi-objective alternatives**
>
> Three of the methods we compare against correspond to multi-objective or Lagrangian reformulations of a loss very similar to the one we consider. The Cali method corresponds to a weighted sum of ECE and the sharpness term for all intervals. Similarly, the Interval method penalizes sharpness and a distance-weighted deviation from zero ECE. QRT enforces calibration on the training data throughout training and minimizes the log-likelihood. Although these methods seemingly tackle the problem in a more direct way, they perform worse than ours. This indicates that the underlying optimization problem is very difficult to solve, most likely due to a large amount of local minima.
>
> Please note that although the pinball loss does not emerge from a Lagrangian formulation of the problem, it does correspond to a loss function that trades off sharpness and calibration, as discussed in Section 2.3. Although rigorously and theoretically quantifying the difference between these methods is difficult, the empirical results suggest that the trade-off obtained by the pinball loss avoids the local minima introduced by Lagrangian-type reformulations. We will make these points clear in the revised manuscript.
>
> **Non-crossing quantiles**
>
> Thank you for this observation. We additionally applied the method of Chernozhukov et al. (2010), a post-processing method that eliminates crossing quantiles, to our approach. We observed a mild improvement in the results:
>
> | Metric | CGQR   Non-crossing  | CGQR |
> |--|--|--|
> | IGD ↓ | 0.25 ± 0.03 |  0.28±0.04  |
> | GD ↓ | 0.16 ± 0.03 | 0.20±0.02  |
> | HV ↑ | 0.95 ± 0.0062 |  0.92 ± 0.01 |
>
> We also note that the approach of neglecting the crossing problem is not uncommon in the literature, as it allows significant flexibility for the resulting model, and as we observed, does not lead to frequent crossings when fully trained (Romano et al., 2019; Zhao et al., 2020; Chung et al., 2021; Kuleshov et al., 2022).
>
> **Guidance for Choosing $\epsilon$**
>
> Theorem 3.2 indicates that, for large values of $\epsilon$, the miscalibration in the worst case will deviate more strongly from $\epsilon$, whereas it is closer to $\epsilon$ for smaller values of $\epsilon$. The empirical results in Fig. 2 show that sharpness with our method improves approximately linearly with miscalibration for miscalibration values above $0.02$. Below that, sharpness increases sharply. This suggests that there are two different regimes of sensitivity for $\epsilon$ ($\epsilon$<0.02 and $\epsilon$>0.02), which can be used to inform $\epsilon$. However, we note that ultimately $\epsilon$ should be chosen with the needs of the application in mind.
>
> Furthermore, we examined how low $\epsilon$ can be chosen before the optimization problem becomes infeasible:
>
> | 10th percentile  | Median | 90th percentile |
> |---|---|---|
> | 0.015   | 0.019   | 0.023   |
>
> For all except the smallest datasets (<2000 points), any $\epsilon$ larger than $0.03$ will yield a feasible result with high probability. We have included these discussions in the revised manuscript.
>
> **Other Uncertainty quantification methods**
>
> The theoretical contribution of our paper is independent of the loss function, hence can be straightforwardly extended to other uncertainty quantification methods. This is also showcased in the experimental section, where we obtain a fair comparison by applying the same selection criterion to different state-of-the-art loss functions. We also note that the QRT method is trained using an NLL loss, a commonly used loss function.
>
> **Comparison with Conformal Prediction**
>
> Conformal prediction methods employ a nonconformity score and a held-out dataset to calibrate a pre-trained model. Similar to our approach, they often aim for marginal calibration. The resulting theoretical results are typically tight, provided the data are exchangeable. Our theoretical assumptions apply a similar mechanism, but are ultimately less tight. This is because we are choosing a model from $m(\epsilon)$ potential models obtained during training, thereby loosening the resulting error bound.
>
> Victor Chernozhukov, Iván Fernández-Val, and Alfred Galichon. Quantile and probability
>  curves without crossing. Econometrica, 2010.
>
> Romano, Yaniv, Evan Patterson, and Emmanuel Candes. "Conformalized quantile regression." Neurips 2019
>
> Zhao, Shengjia, Tengyu Ma, and Stefano Ermon. "Individual calibration with randomized forecasting." ICML 2020.
>
> Chung, Youngseog, Willie Neiswanger, Ian Char, and Jeff Schneider. "Beyond pinball loss: Quantile methods for calibrated uncertainty quantification." Neurips 2021
>
> Kuleshov, Volodymyr, and Shachi Deshpande. "Calibrated and sharp uncertainties in deep learning via density estimation." In ICML 2022.

---

### Meta-Review · Area_Chair_SqUB · 2026-01-12

**Summary:**

This paper proposes Calibration-Guided Quantile Regression, an approach that iteratively adjusts quantile regression models using calibration diagnostics with the goal of improving predictive calibration. The motivation is relevant, and the paper explores an interesting direction at the intersection of quantile estimation and uncertainty calibration. The empirical results suggest that the method can improve calibration metrics in some settings.

However, after considering the reviews and the rebuttal, I do not believe the paper meets the acceptance bar for ICLR. A key concern is whether the proposed procedure offers a principled improvement over existing quantile regression and calibration techniques, or whether it risks overfitting calibration diagnostics without providing stronger statistical guarantees. Several reviewers questioned the conceptual framing, novelty, and validity of the method, and these concerns were not fully resolved by the rebuttal. While the rebuttal clarifies intent and implementation details, it does not sufficiently address doubts about the methodological soundness and broader applicability of the approach. Overall, the contribution appears incremental and potentially problematic in its current form, leading to a rejection recommendation.

**Reviewer Concerns:**

Some concerns were addressed by the rebuttal. ALqY initially raised issues stemming from confusion about the method and, after clarification, viewed the contribution more positively. NdLJ requested clearer explanation of the algorithmic steps, assumptions, and empirical setup, and the rebuttal improved clarity on these points.

However, important concerns remain outstanding. t1Nn raised fundamental objections regarding the conceptual validity of using calibration feedback to guide model fitting, arguing that this may introduce circularity and undermine statistical guarantees; these concerns were not alleviated. WBvh questioned the degree of novelty and whether the observed improvements meaningfully exceed what can be achieved with existing calibration or quantile regression techniques, and this concern also remains. More broadly, uncertainty persists about whether the method provides reliable benefits beyond optimizing calibration metrics on the observed data.

**Reviewer Scores:**

t1Nn (rating 0).
This reviewer expressed principled objections to the core methodology. The rebuttal does not resolve these concerns, and the rating would likely remain unchanged.

WBvh (rating 2).
This reviewer was unconvinced about novelty and contribution strength. The rebuttal is unlikely to change this assessment.

NdLJ (rating 4).
This reviewer raised concerns primarily about clarity and empirical justification. While the rebuttal improves clarity, the overall assessment would likely remain similar.

ALqY (rating 6).
This reviewer ultimately viewed the paper favorably after clarification. The rating would likely remain unchanged.

Overall, while the rebuttal improved exposition, it did not materially shift reviewer opinions regarding the core methodological concerns, and significant rating changes are unlikely.

---

### Decision · Program_Chairs · 2026-01-26

Reject